**From the middle stratosphere to the surface, using nitrous oxide to constrain the stratosphere-troposphere exchange of ozone**

Daniel J. Ruiz[1] and Michael J. Prather[1]

[1]Department of Earth System Science, University of California, Irvine, CA 92697-3100, USA

*Correspondence to:* Daniel J. Ruiz (djruiz@uci.edu)

**Abstract**

Stratosphere-troposphere exchange (STE) is an important source of tropospheric ozone, affecting all of atmospheric chemistry, climate, and air quality. The study of impacts needs STE fluxes to be resolved by latitude and month, and for this we rely on global chemistry models, whose results diverge greatly. Overall, we lack guidance from model-measurement metrics that inform us about processes and patterns related to the STE flux of ozone ($O_3$). In this work, we use modeled tracers ($N_2O$, $CFCl_3$) whose distributions and budgets can be constrained by satellite and surface observations, allowing us to follow stratospheric signals across the tropopause. The satellite derived photochemical loss of $N_2O$ on annual and quasi-biennial cycles can be matched by the models. The STE flux of $N_2O$-depleted air in our chemistry transport model drives surface variability that closely matches observed fluctuations on both annual and quasi-biennial cycles, confirming the modeled flux. The observed tracer correlations between $N_2O$ and $O_3$ in the lowermost stratosphere provide a hemispheric scaling of the $N_2O$ STE flux to that of $O_3$. For $N_2O$ and $CFCl_3$, we model greater southern hemispheric STE fluxes, a result supported by some metrics, but counter to prevailing theory of wave-driven stratospheric circulation. The STE flux of $O_3$, however, is predominantly northern hemispheric, but evidence shows that this is caused by the Antarctic ozone hole reducing southern hemispheric $O_3$ STE by 14%. Our best estimate of the current STE $O_3$ flux based on a range of constraints is 400 Tg($O_3$)/yr with a one-sigma uncertainty of ±15% and with a NH:SH ratio ranging from 50:50 to 60:40. We identify a range of observational metrics that can better constrain the modeled STE $O_3$ flux in future assessments.

## 1. Introduction & Background

The influx of stratospheric ozone ($O_3$) into the troposphere affects its distribution, variability, lifetime, and thus its role in driving climate change and surface air pollution (Zeng et al., 2010; Hess et al., 2015; Williams et al., 2019). The net stratosphere-to-troposphere exchange (STE) flux of $O_3$ has a regular seasonal cycle in each hemisphere that is an important part of the tropospheric $O_3$ budget (Stohl et al., 2003). Such fluxes are not directly observable, and we rely on observational estimates using trace-gas ratios, in particular the $O_3$:$N_2O$ ratio in the lower stratosphere (Murphy and Fahey, 1994; McLinden et al., 2000), or dynamical calculations using measured/modeled winds and $O_3$ abundances (Gettelman et al., 1997; Olsen et al., 2004; Yang et al., 2016). The uncertainty in these estimates does not effectively constrain the wide range found in the models being used to project future ozone (Young et al., 2013, 2018; Griffiths et al., 2021). Here we present the case for using the observed variations in nitrous oxide ($N_2O$) from the middle stratosphere to the surface in order to constrain the STE flux of $O_3$. A similar case

has been made for the radionuclide $^7$Be (Liu et al., 2016), but N$_2$O has a wealth of model-
observation metrics on hemispheric, seasonal, and interannual scales that constrains its STE flux
very well (Prather et al., 2015; Ruiz et al., 2021).
Ozone-rich stratospheric air has been photochemically aged and is depleted in trace gases such as
N$_2$O and chlorofluorocarbons (CFCs).  For these trace gases, the overall circulation from
tropospheric sources to stratospheric destruction and back is part of the lifecycle that maintains
their global abundance (Holton, 1990).  For N$_2$O and CFCs, this cycle of (i) loss in the middle to
upper stratosphere, (ii) transport to the lowermost stratosphere (Holton et al., 1995), and then (iii)
influx into the troposphere produces surface variations not related to surface emissions
(Hamilton and Fan, 2000; Nevison et al., 2004; Hirsch et al., 2006; Montzka et al., 2018; Ray et
al., 2020; Ruiz et al., 2021).  In this work we relate our modeled STE fluxes to variations at the
surface and throughout the stratosphere, linking the fluxes of N$_2$O to O$_3$ through stratospheric
measurements.  Our goal is to develop a set of model metrics founded on observations that are
related to the STE O$_3$ flux and can be used with an ensemble of models to determine a better,
constrained estimate for the flux, including seasonal, interannual, and hemispheric patterns.  This
approach is similar to efforts involving the ozone depletion recovery time (Strahan et al.,
2011) and projections of future warming (Liang et al., 2020; Tokarska et al., 2020).
In a previous work (Ruiz et al., 2021, hence R2021) we showed that historical simulations with
three chemistry transport models (CTMs) were able to match the interannual surface variations
observed in N$_2$O.  These were clearly driven by the stratospheric quasi-biennial oscillation
(QBO) which appears to be the major interannual signal in stratospheric circulation and STE
(Kinnersley and Tung, 1999; Baldwin et al., 2001; Olsen et al., 2019).  In this work, we calculate
the monthly latitudinal STE fluxes of O$_3$, N$_2$O, and CFCl$_3$ (F11), establish a coherent picture
relating fluxes to observed abundances, and summarize the methods in Section 2.  In section 3,
we examine the annual and interannual cycles as well as geographic patterns of modeled STE
flux.  In section 4, we relate the surface variability of N$_2$O to its STE flux.  We find some
evidence to support our model result that the STE flux of depleted-N$_2$O air is greater in the
southern hemisphere than in the northern, thus altering the asymmetry in surface emissions in the
source inversions (Nevison et al., 2007; Thompson et al., 2014).  In section 5, we examine the
lowermost stratosphere to understand the large north-south asymmetry found in O$_3$ STE versus
N$_2$O or F11 STE, and find a clear signal of the Antarctic ozone hole in STE.  In section 6, we
examine the consistency of the model calculations of STE flux and derive a best estimate for the
O$_3$ flux from this and previous studies.  We summarize a sequence of model metrics, primarily
using O$_3$ and N$_2$O, that can narrow the range in the tropospheric O$_3$ budget terms for the multi-
model intercomparison projects used in tropospheric chemistry and climate assessments.
**2. Methods**
The modeled STE fluxes here are calculated with the UCI (University of California Irvine) CTM
driven by 3-hour forecast fields from the European Centre for Medium-range Weather Forecasts
(ECMWF) Integrated Forecast System (IFS Cycle 38r1 T159L60) for years 1990-2017, as are
the calculations in R2021.  The CTM uses the IFS native 160x320 Gauss grid (~1.1°) with 60
layers, about 35 of which are in the troposphere.  The stratospheric chemistry uses the linearized
model Linoz v3 and includes O$_3$, N$_2$O, NO$_y$, CH$_4$, and F11 as transported trace gases (Hsu and
Prather, 2010; Prather et al., 2015; Ruiz et al., 2021).  There is no tropospheric chemistry, but
rather a boundary-layer e-fold to a specified abundance, or a surface boundary reset to an
abundance.  Equivalent effective stratospheric chlorine levels are high enough to drive an
Antarctic ozone hole, which is observed throughout this period.  Thus, the ozone-hole chemistry
in Linoz v3 is activated for all years, and the amount of $O_3$ depleted depends on the Antarctic
meteorology of that year (Hsu and Prather, 2010).
The STE flux is calculated using the e90 definition of tropospheric grid cells (Prather et al.,
2011) and the change in tropospheric tracer mass from before to after each tracer transport step
as developed at UCI (Hsu et al., 2005; Hsu and Prather, 2009; Hsu and Prather, 2014).  This
method is precise and geographically accurate for $O_3$ and is self-consistent with a CTM's tracer-
transport calculation (Tang et al., 2013; Hsu and Prather, 2014).  Extensive comparisons with
other methods of calculating STE are shown in Hsu and Prather (2014).  Annual-mean STE
fluxes are calculated from the full 28-year (336 month) time series as 12-month running means,
and the annual cycle of monthly fluxes is the average of the 28 values for each month.
R2021 modeled the surface signal of stratospheric loss with the decaying tracers, N2OX and
F11X (e.g., (Hamilton and Fan, 2000; Hirsch et al., 2006). These X-tracers have the identical
stratospheric chemical loss frequencies as $N_2O$ and $CFCl_3$, respectively, but no surface sources
and are therefore affected only by the stratospheric sink and atmospheric transport.  The multi-
decade (F11X) to century (N2OX) decays are easily rescaled using a 12-month smoothing filter
to give stationary results and a tropospheric mean abundance of 320 ppb.  We treat F11X like
N2OX with the same initial conditions and molecular weight.  Budgets for N2OX are reported,
as in $N_2O$ studies (Tian et al., 2020), as Tg of N as $N_2O$.  These rescaled N2OX and F11X tracers
are designated simply as N2O (not $N_2O$) and F11.  Our F11 STE fluxes are thus unrealistically
large compared to current $CFCl_3$ fluxes, but can be easily compared with our N2O results.
When trying to calculate the STE flux of $N_2O$-depleted air across the tropopause, we found that
the Hsu method was numerically noisy because the gradient across the tropopause, unlike that of
$O_3$, was negligible.  Thus, for this work we created the complementary tracers cN2OX and
cF11X:  for each kg of the X-tracer (i.e., N2OX) destroyed by photochemistry, 1 kg of its
complementary tracer (cN2OX) is created. Air parcels that are depleted in N2OX (F11X) are
therefore rich in cN2OX (cF11X).  After crossing the tropopause, cN2OX and cF11X are
removed through rapid uptake in the boundary layer, thus creating sharp gradients at the
tropopause in parallel with that of $O_3$.  As a check, we compared the boundary layer sinks of the
c-tracers with their e90-derived STE fluxes and find that their sums are identical.  The c-tracers
and their STE fluxes are rescaled as are the X-tracers to give them a stationary time series
corresponding to a tropospheric abundance of 320 ppb for their parallel X tracers. We designate
these scaled tracers simply as cN2O and cF11.
**3.  Modeled STE fluxes**
*3.1 Global and hemispheric means*
The 28-year mean of global $O_3$ STE is 390±16 Tg/yr (positive flux means stratosphere to
troposphere, the ± values are the standard deviation of the 28 annual means and do not represent
uncertainty). This value is well within the uncertainty in the observation-based estimates
(Murphy and Fahey, 1994; Olsen et al., 2001), and far from the extreme ranges of the 34 models
in the latest Tropospheric Ozone Assessment Report (Young et al., 2018), 150 to 940 Tg/yr. The
global STE flux of cN2O is 11.5±0.7 Tg/yr, and that of cF11 is 23.5±1.5 Tg/yr.  These fluxes for
cN2O and cF11 match the total long-term troposphere-to-stratosphere flux of N2O and F11 as
derived from their stratospheric losses.  The cF11 budget is about twice as large as cN2O,
because F11 is photolyzed rapidly in the lower-middle stratosphere (~24 km) instead of the
upper stratosphere like N2O (~32 km).  The seasonal mean pattern of STE fluxes are shown in
Figure 1.  The large majority of STE flux enters the troposphere at 25°-45° latitude in each
hemisphere, but there is a broadening of the northern flux to 65°N in Jun-Jul.  The importance of
this region about the sub-tropical jet for STE is supported by satellite data where stratospheric
folding events (high $O_3$ in the upper troposphere) are found at the bends of the jet (Tang and
Prather, 2010).
Given the small STE fluxes in the core tropics, the northern hemisphere (NH) and southern
hemisphere (SH) fluxes are distinct.  The annual mean of NH $O_3$ STE is 208±11 Tg/yr and is
slightly larger than the SH mean of 182±11 Tg/yr.  This NH:SH ratio of 53:47 is typically found
in other studies (Gettelman et al., 1997; Hsu and Prather, 2009; Yang et al., 2016), although
some have higher ratios like 58:42 (Hegglin and Shepherd, 2009; Meul et al., 2018).  In contrast,
for cN2O and cF11, the NH flux (5.1±0.4 Tg/yr and 10.6±0.8 Tg/yr, respectively) is smaller than
the SH flux (6.4±0.5 Tg/yr and 12.9±1.0 Tg/yr, respectively), giving a NH:SH ratio of about
45:55.  The established view on STE is that the flux is wave-driven and under downward control,
and thus the NH flux is much greater than the SH flux (see Table 1 of Holton et al., 1995; also
Appenzeller et al., 1996).  Our unexpected results require further analysis including evidence for
hemispheric asymmetry in observations which is shown in section 4 along with other model
metrics.

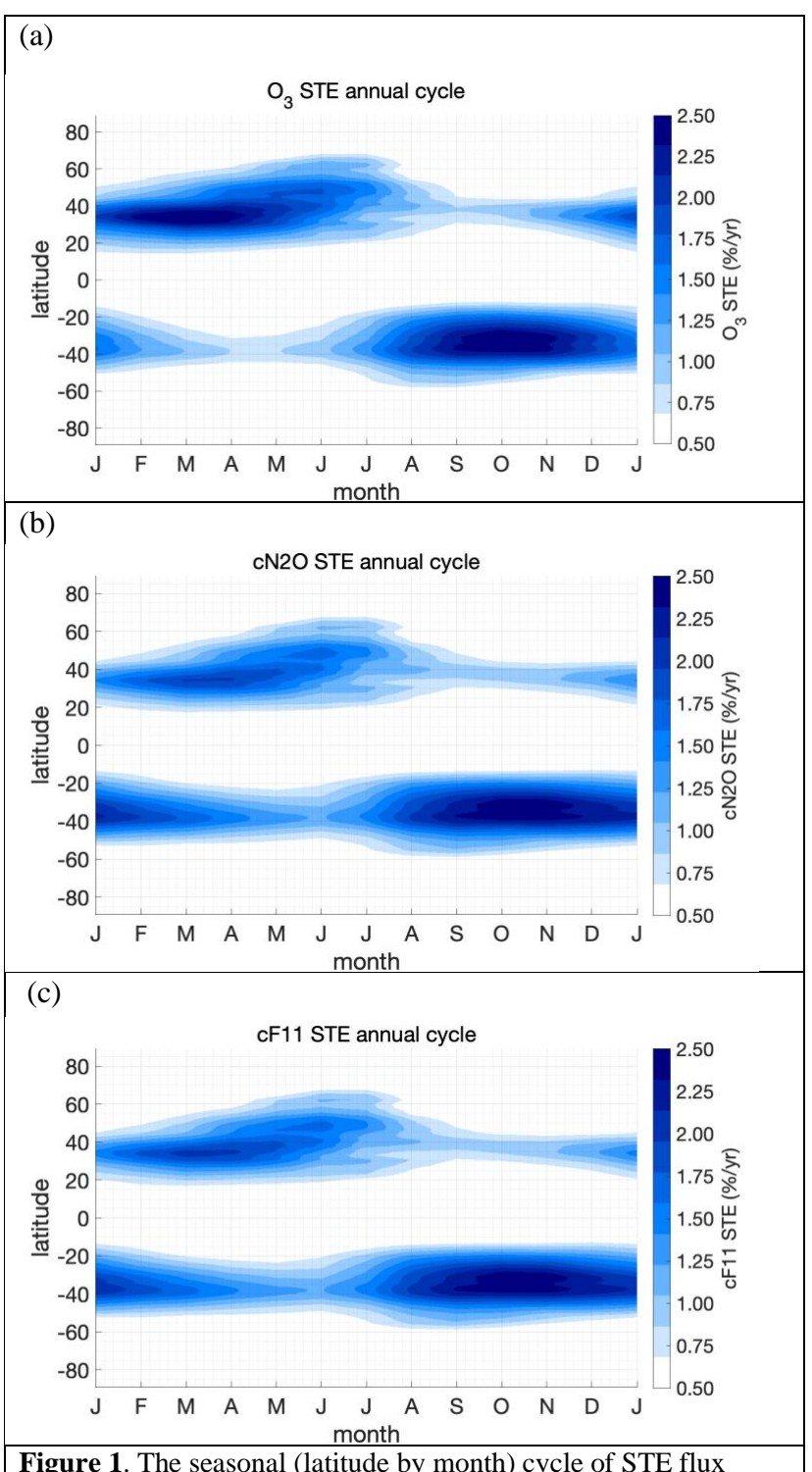

**Figure 1**. The seasonal (latitude by month) cycle of STE flux (Tg/yr) for **(a)** $O_3$, **(b)** cN2O, and **(c)** cF11. Each month is averaged for years 1990-2017 (e.g., the 28 Januarys are averaged). The colorbar units are % of global, annual mean STE in each bin (1 month by ~1.1° latitude).


*3.2 Seasonal cycle*
The seasonal cycles of STE fluxes summed over global, NH, and SH are shown in Figure 2. The
scales are given as the annual rate (as if the monthly rate were maintained for the year), and each
species has a different axis. The right y-axes are kept at a N2O:F11 ratio of 1:2. Despite large
differences in the stratospheric chemistry across all three species, the seasonal cycle of STE is
highly correlated (Pearson's correlation coefficient cc > 0.98, except for SH $O_3$), indicating that
all three enter the troposphere from a seasonally near-uniform mixture of $O_3$:N2O:F11 in the
lowermost stratosphere.
Global STE peaks in June and reaches a minimum in November. The two hemispheres have
dramatically different seasonal amplitudes and somewhat opposite phases. NH peak STE for all
3 species occurs in the late boreal spring (May-June), while that in the SH occurs at the start of
austral spring (September-October). In the NH $O_3$ STE peaks a month before the c-tracers, and
in the SH the whole annual cycle of $O_3$ is shifted a month earlier. The NH STE seasonal
amplitude is very large for all species (~ 4:1 ratio max-to-min) with exchange almost ceasing in
the fall. In contrast, the SH STE is more uniform year-round with a 1.5:1 ratio for cN2O and
cF11, and 2.2:1 for $O_3$. Other models with similar NH and SH $O_3$ fluxes show different seasonal
amplitudes and phasing (see Fig. 6 of Tang et al., 2021), which will affect tropospheric $O_3$
abundances. It is important to develop observational metrics that test the seasonality of the
lowermost stratosphere related to STE fluxes, and to establish monthly STE fluxes as a standard
model diagnostic.
An interesting result here is the very tight correlation of the monthly cN2O and cF11 STE while
the $O_3$ STE is sometimes shifted. Loss of N2O and F11 occurs at very different altitudes in the
tropical stratosphere (~32 km and ~24 km, respectively), but both have similar seasonality in
loss, driven mostly by the intensity of sunlight along the Earth's orbit (N2O loss peaks in Feb and
reaches a minimum in July, see Fig. 4 from Prather et al. (2015). Photochemical losses of N2O
and F11 drop quickly for air descending from the altitudes of peak loss in the tropics and hence
the relative cN2O and cF11 STE fluxes are locked in. $O_3$, however, continues to evolve
photochemically from 24 km to 16 km (upper boundary of the lowermost stratosphere), through
net photochemical production in the tropics and loss at mid- and high-latitudes that depends on
sunlight and is thus seasonal. There may be observational evidence for the patterns modeled
here in the correlation of these three tracers in the lower (16-20 km) and lowermost (12-16 km)
extratropical stratosphere (see section 4).

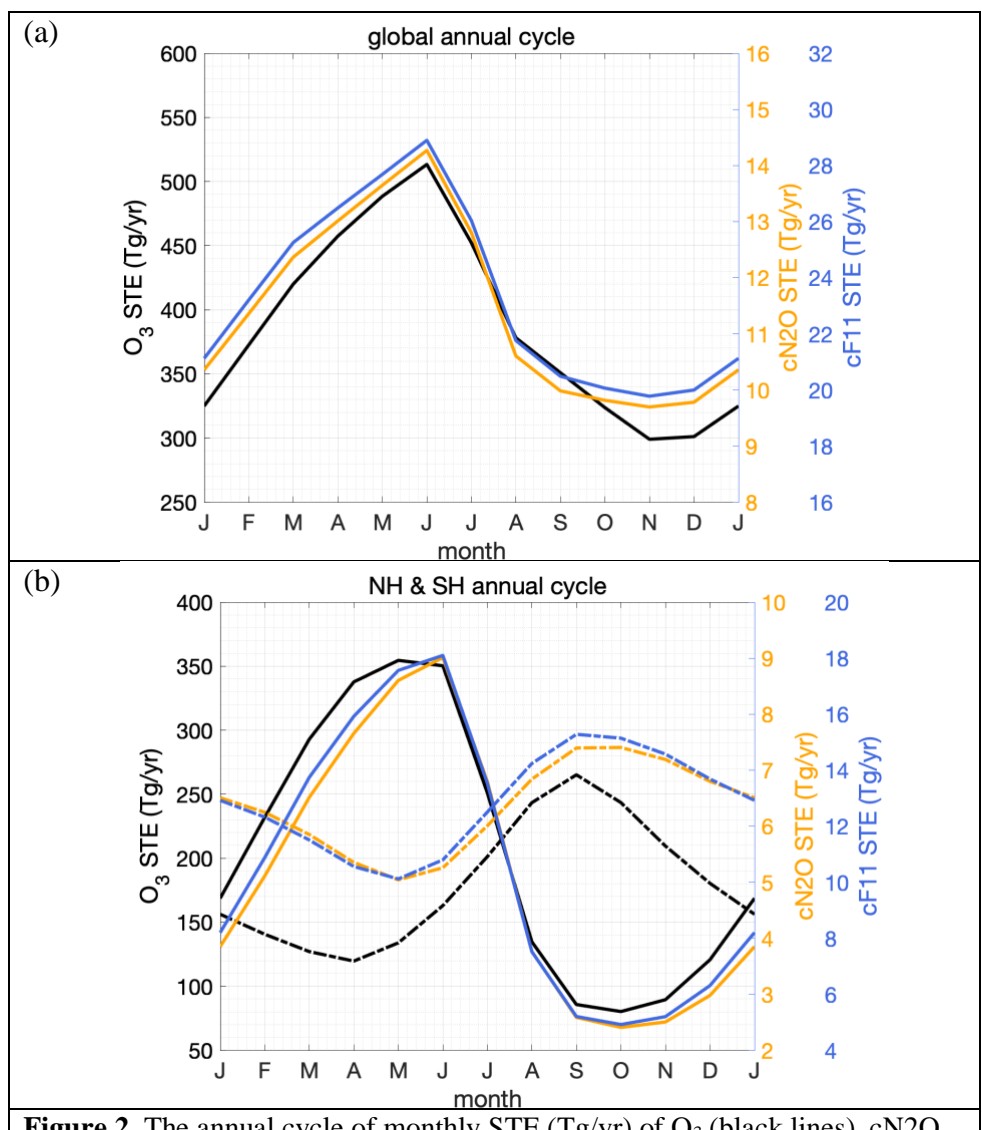

**Figure 2**. The annual cycle of monthly STE (Tg/yr) of $O_3$ (black lines), cN2O (orange lines), and cF11 (blue lines). **(a)** Global STE fluxes, and **(b)** hemispheric STE fluxes (NH, solid lines; SH, dashed lines). Each month is averaged for years 1990-2017 (e.g., the 28 Januarys are averaged). Note the different y-axes for each tracer in each panel.

### 3.3 Interannual variability

Interannual variability (IAV) of $N_2O$ loss and its lifetime is associated primarily with the QBO (most recently, R2021). When the QBO is in its easterly (westerly) phase the entire overturning circulation is enhanced (suppressed) (Baldwin et al., 2001). This results in more (less) air rich in $N_2O$ and F11 being transported from the troposphere to the lower or middle stratosphere, thereby increasing (decreasing) the $N_2O$ and F11 sinks (Prather et al., 2015; Strahan et al., 2015). From the tropical stratosphere, the overturning circulation transports air depleted in $N_2O$ and F11 into the lowermost extratropical stratosphere, where it enters the troposphere. R2021 showed that the

observed surface variability of $N_2O$ from this circulation can be modeled and has a clear QBO
signal, but one that is not strongly correlated with the QBO signal in stratospheric loss.
We generate the IAV of STE fluxes for $O_3$, cN2O, and cF11 in Figure 3abc with panels for
global, NH, and SH. Values are 12-month running means, and so the first modeled point at
1990.5 is the sum of STE for Jan through Dec of 1990. In Figures 3bc, we also show the
seasonal amplitude of STE with double-headed arrows on the left ($O_3$) and right (cN2O and
cF11). In a surprising result, the large NH-SH differences in seasonal amplitude are not reflected
in the IAV where NH and SH amplitudes are similar for all three tracers. The QBO modulation
of the lowermost stratosphere and STE appears to be unrelated to the seasonal cycle in STE.
Global STE for all three tracers shows QBO-like cycling throughout the 1990-2017 time series:
cN2O and cF11 are well correlated (cc ~ 0.9), but either species with $O_3$ is much less so (cc <
0.7). The hemispheric breakdown provides key information regarding $O_3$. In the NH the STE
IAV is similar across all three tracers with high correlation coefficients (cc = 0.82 for $O_3$-cN2O,
0.83 for $O_3$-cF11, and 0.94 for cN2O-cF11). Conversely in the SH, $O_3$ STE diverges from the c-
tracer fluxes, showing opposite-sign peaks in 2003 and 2016. The corresponding SH
correlations are (cc = 0.38, 0.65. 0.85). The loss of correlation between cN2O and cF11 is
unusual: cN2O STE drifts downward relative to cF11 STE, particularly after 2007; nevertheless,
the fine structure after 2007 is well matched in both tracers.
In the SH, the massive loss of $O_3$ within the Antarctic vortex, when mixed with the extra-polar
lowermost stratosphere will systematically shift the $O_3$ STE to lower values, with less impact on
the cN2O and cF11 STE. The IAV of the Antarctic winter vortex, in terms of the amount of $O_3$
that is depleted (see Fig. 4-4 of WMO, 2018), appears to drive the decorrelation of the SH STE
fluxes and is analyzed in section 4.
In the NH, the high variability of the Arctic winter stratosphere can modulate the total $O_3$ STE
flux (e.g., Hsu and Prather, 2009) but appears to maintain the same relative ratio with the cN2O
and cF11 fluxes. Model results here indicate that in the NH, the IAV of $O_3$, cN2O, and cF11 STE
fluxes are synchronized, and thus the air masses entering the lowermost stratosphere have the
same chemical mixtures from year to year. We know that cold-temperature activation of
halogen-driven $O_3$ depletion in the Arctic winter at altitudes above 400 K (potential temperature)
can produce large IAV in column ozone (Manney et al., 2011); but the magnitude is still much
smaller than in the Antarctic; and it may not reach into the lowermost stratosphere (<380K
potential temperature). This model accurately simulates Antarctic $O_3$ loss (section 4), but we
have not evaluated it for Arctic loss, and the Arctic conditions operate closer to the thresholds
initiating loss where Linoz v3 chemistry may be inadequate. The same meteorology and
transport model with full stratospheric chemistry is able to simulate Arctic $O_3$ loss (Oslo's
CTM2: Isaksen et al., 2012), and thus it will be possible to re-evaluate the NH IAV with such
models or with lowermost stratosphere tracer measurements.

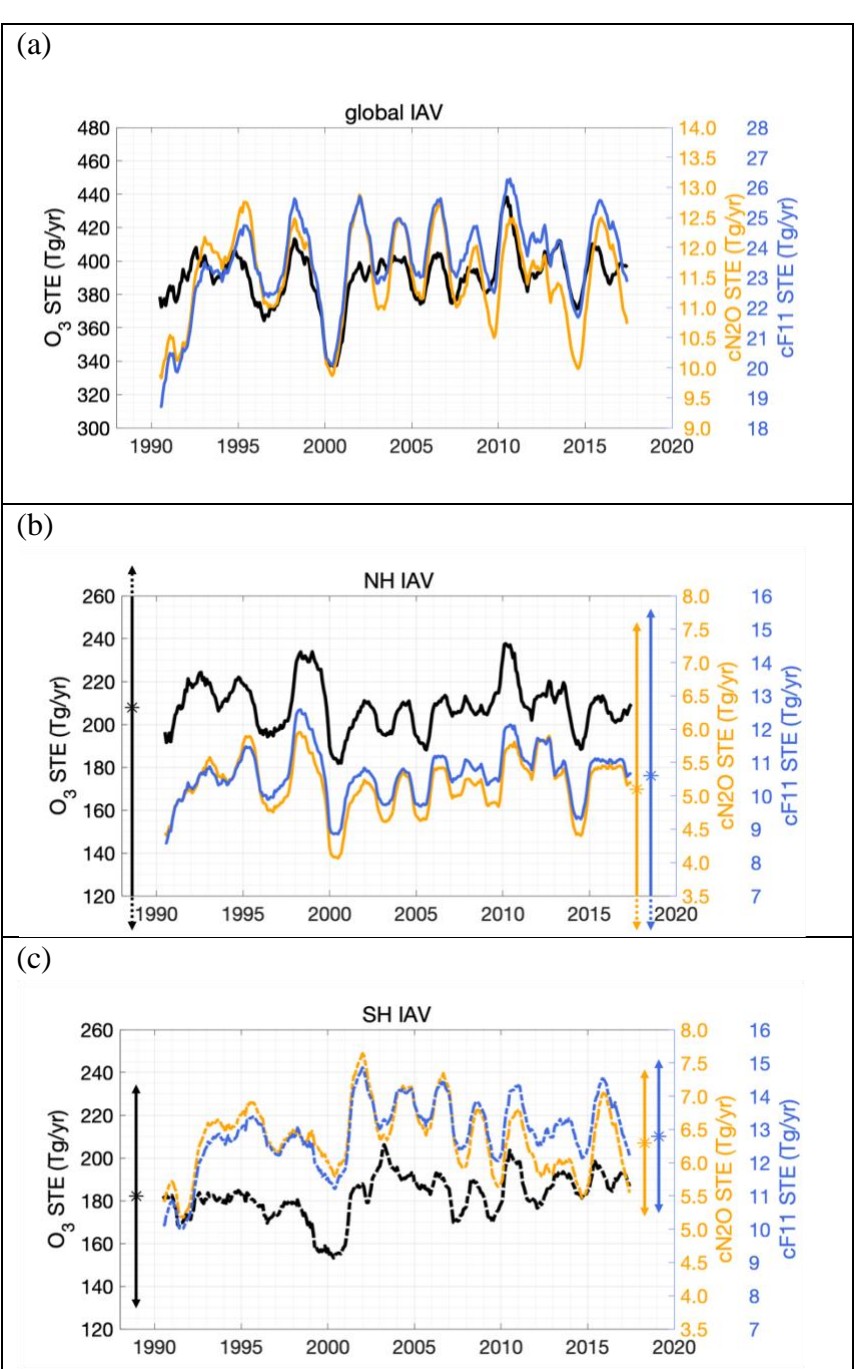

**Figure 3.** **(a)** Global STE (Tg/yr), calculated at e90 tropopause, of O$_3$ (black line; left y-axis), cN2O (orange line; orange right y-axis), and cF11 (blue line; blue right y-axis) for years 1990-2017. Values are 12-month running means, and so the first point at 1990.5 is the sum of STE for Jan through Dec of 1990. **(b)** NH STE. **(c)** SH STE. The scales for cN2O and cF11 are kept in a 1:2 ratio. The asterisks and vertical double-headed arrows (**b** & **c**) depict the seasonal mean and amplitude for each species in each hemisphere.


*3.4 The link from stratospheric loss to STE flux*
What is unusual about the very tight correlation of cN2O and cF11 STE fluxes is that the
photochemical loss of N2O and F11 occurs at very different altitudes in the tropical stratosphere,
which are not in phase with respect to the QBO as shown in R2021 (their Fig. 2). The separate
phasing of cN2O and cF11 production is lost, presumably by diffusive tracer transport, by the
time they reach the extratropical lowermost stratosphere. The overall synchronization of the
STE fluxes implies that the absolute STE flux is driven primarily by variations in venting of the
lowermost stratosphere as expected (Holton et al., 1995; Appenzeller et al., 1996) rather than by
variations in the chemistry of the middle stratosphere.
This disconnect between the chemical signals generated by the prominent QBO signature of
wind reversals, upwelling in the tropical stratosphere, and the STE fluxes is also clear in the
magnitude of the loss versus STE. For N2O, the IAV of cN2O production has a range of ±0.5
Tg/yr, whether from the Aura Microwave Limb Sounder (Aura-MLS) observations or the model;
whereas the IAV of cN2O STE flux is ±1.1 Tg/yr. The same is true in relative terms for cF11.
Thus, the modulation of the lowermost stratosphere by the QBO is clearly a part of the overall
changes in stratospheric circulation related to the QBO (Tung and Yang, 1994a; Kinnersley and
Tung, 1999) and is the dominant source of IAV for these three greenhouse gases.
*3.5. The QBO signal*
To examine the QBO cycle in STE flux, we build a composite pattern (see R2021, Fig. 3 of $N_2O$
surface variations), by synchronizing the STE IAV in Figure 2 with the QBO cycle. The sync
point (offset = 0 months) is taken from one of the standard definitions of the QBO phase change,
i.e., the shift in sign of the 40-hPa tropical zonal wind from easterly to westerly (Newman,
2020). The 1990-2017 model period has 12 QBO cycles, but we restrict our analysis here to
years 2001-2016 to overlap with the observed surface N2O data. This period includes seven
QBO phase transitions (01/2002, 03/2004, 04/2006, 04/2008, 08/2010, 04/2013, 07/2015), but
the observed surface N2O is highly anomalous during the QBO centered on 08/2010 (R2021), so
we remove it from our comparison for consistency with R2021 (see their Fig S4d). The resulting
QBO composites for NH and SH in Figure 4 span 28 months.
In the NH, the QBO modulation of all three tracers is similar: STE flux begins to increase at an
offset of -8 months and continues to increase slowly for a year, peaking at an offset of +4
months; thereafter it decreases more rapidly in about ½ year (offset = +10). The rise-and-fall
cycle takes about 18 months. In the SH, the pattern for cN2O and cF11 is more sinusoidal and is
shifted later by ~3 months. The SH amplitude of the c-tracers is slightly larger relative to the
hemispheric mean flux than in the NH, and thus the SH QBO signal is larger than the NH by
about 40%. Thus, over the typical QBO cycle centered on the sync point, more depleted $N_2O$
and F11 is entering the SH than in the NH. For $O_3$, the SH modulation of STE is irregular and
reduced compared with the NH. Our hypothesis here, consistent with the annual cycle of STE
(Figure 1), is that the breakup of the Antarctic ozone hole has a major impact on STE,
particularly that of $O_3$, and that its signal has large IAV that does not synchronize with the QBO.
Surprisingly, the large wintertime IAV in the NH Arctic, in the form of sudden stratospheric
warmings, does not seem to have a major role in STE fluxes as noted above. This model may
miss some of the Arctic O₃ depletion, but it accurately simulates the warmings, which must have
a small impact on STE because they do not disrupt the clear QBO signal in the c-tracers.

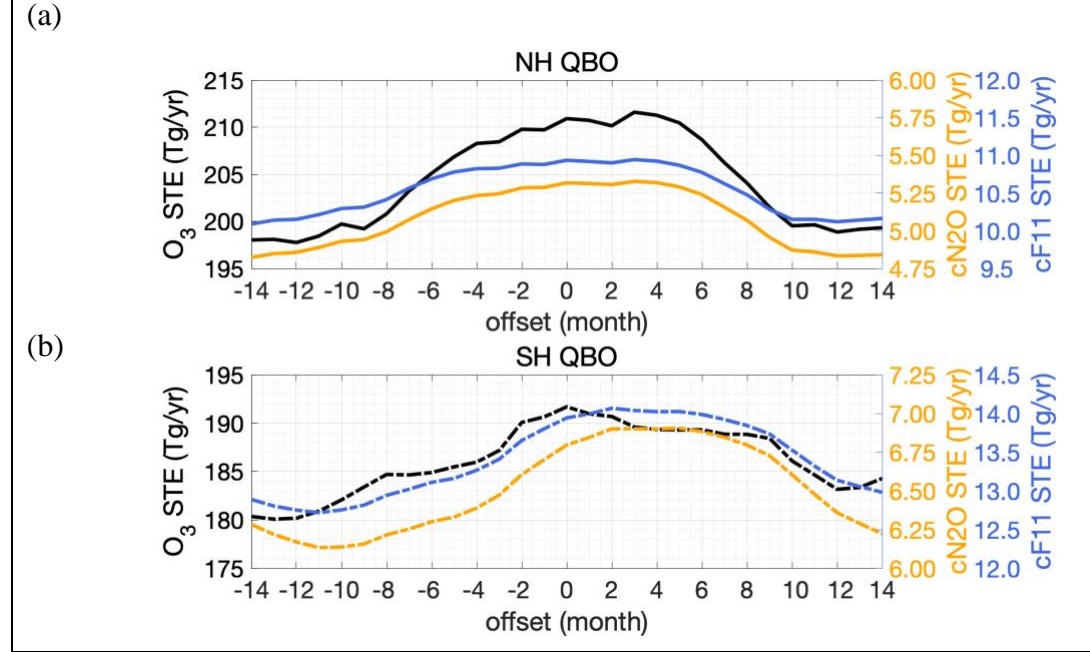

**Figure 4.** QBO composites of the STE of O$_3$ (black lines; left y-axes), cN2OX (orange lines; orange right y-axes), and cF11X (blue lines; blue right y-axes) for the **(a)** NH (0°-90°N; solid lines) and **(b)** SH (0°-90°S; dashed lines). These composites are averages centered on the QBO phase transition at 40 hPa throughout the period of surface observations (years 2001-2016, excluding the 08/2010 observed anomaly, for a total of 6 QBOs). Note: the y-axes limits are different for each panel, but the interval scale is consistent for each tracer.

**4. Surface variability of N$_2$O related to STE flux**
Surface variability of N$_2$O is driven by surface emissions, stratospheric loss, and atmospheric
transport that mixes the first two signals. R2021 explored the variability originating only from
stratospheric chemistry using the decaying tracer N2OX. Here, we use surf-N2O to denote the
surface abundances of N2OX when corrected to steady state. R2021 showed that three
independent chemistry-transport models produced annual and QBO patterns in surface N$_2$O
simply from stratospheric loss. In this paper we link surf-N2O to the STE cN2O flux, which is
linked above to the STE O$_3$ flux.
The observed surface N$_2$O, denoted obs-N$_2$O and taken from the NOAA network (Dlugokencky
et al., 2019), shows a slowly increasing abundance (~0.9 ppb/yr) with a clear signal of annual
and interannual variability at some latitudes (see R2021). We calculate annual and QBO-
composite obs-N$_2$O after de-trending and restrict analysis in this section to model years 2001-
2016 to be consistent with the surface data. The latitude-by-month pattern of obs-N$_2$O includes
the impact of both stratospheric loss (~13.5 Tg/yr) and surface emissions (~17 TgN/yr), with the
preponderance of emissions being in the NH (Tian et al., 2020). Total emissions are not
expected to have large IAV but may have a seasonal cycle. The seasonal variation of surface
N$_2$O can also be driven by seasonality in the interhemispheric mixing of the NH-SH gradient (~1
ppb).
*4.1 Annual cycle*
Figure 5 replots the hemispheric mean annual cycles of cN2O STE flux alongside the annual
cycles of surf-N2O and obs-N$_2$O. As noted above, the STE in each hemisphere is almost in
opposite phase, as is the modeled surf-N2O (taken from Fig. 5 of R2021). The NH:SH
amplitude ratio is about 2.4:1 for both STE and surf-N2O. The lag from peak STE flux of cN2O
(negative N$_2$O) to minimum surf-N2O is about 3 months. Such a 90° phase shift is expected for
the seasonal variation of a long-lived tracer relative to a seasonal source or sink. The time lag
between the signal at the tropopause and at the surface, the tropospheric turnover time, should be
no more than a month. Surprisingly, the cN2O STE seasonal amplitude is much larger in the NH
(±3.4 Tg/yr) than in the SH (±1.3 Tg/yr), although the SH mean (6.5 Tg/yr) is larger than the NH
(5.2 Tg/yr). Essentially, there is more variability of air depleted in N$_2$O entering the NH, but air
entering the SH has a larger overall deficit. Thus in our model, the stratosphere creates a NH-SH
gradient of +0.3 ppb at the surface, which is a significant fraction of the observed N-S difference
of +1.3 ppb (R2021). This important result needs to be verified with other models or analyses
because it constrains the NH-SH location of sources.
In the NH, as noted in R2021, the two surface abundances, surf-N2O and obs-N$_2$O, have the
same amplitude and phase, implying that, if the model is correct, the emissions-driven surface
signal has no seasonality, although we know that some important emissions are seasonal
(Butterbach-Bahl et al., 2013). In the SH, the surf-N2O signal is much smaller, in parallel with
the small seasonal amplitude in cN2O STE, but it is out of phase with the obs-N$_2$O. This result
implies that the SH has some highly seasonal sources, or simply that the forcing of SH surf-N2O
by the seasonal cycle of cN2O is weak. Indeed, this is what we might expect from Figure 3: In
the NH the seasonal amplitude in N$_2$O overwhelms the IAV amplitude and is driving the obs-
N$_2$O; but in the SH, both amplitudes are comparable. Given the quasi-regular nature of the
QBO, it would interfere with the seasonal cycle and likely change its phase (as found for other
models in R2021).
In the NH, the annual cycle of O$_3$ and cN2O STE are clearly linked. If we accept that the obs-
N2O NH seasonal cycle is simply driven by the STE flux, then how will tropospheric O$_3$ respond
seasonally? A mole-fraction scaling of the STE fluxes gives an O$_3$:N$_2$O ratio of ~25, and thus
scaling the surf-N2O amplitude gives a large O$_3$ surface seasonality of ~18 ppb. However, the
residence time of a tropospheric O$_3$ perturbation is ~1 month, and thus the peak surface
abundance will lag the peak STE flux by only about a month and not by 3 months as for N$_2$O.
O$_3$ will equilibrate with the flux on monthly timescales and not accumulate. Thus, our estimate
is that NH 30°-90° surface ozone might increase about 5 ppb, peaking in June, due to the STE
flux. In the SH, seasonal patterns are weaker and not well defined, and thus no obvious STE O$_3$
signal is expected.

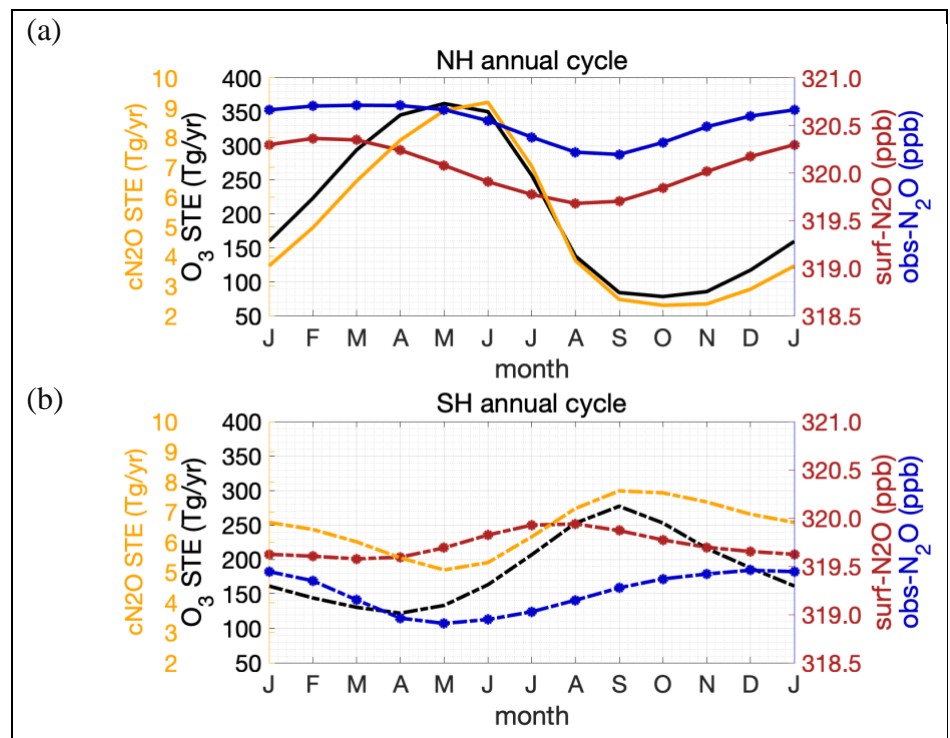

**Figure 5.** The annual cycle of $O_3$ and cN2O STE (black and orange lines; left y-axes), and the surf-N2O and obs-$N_2O$ (red and blue knotted lines; right y-axes) taken from R2021 (see their figure 5) for the **(a)** NH and **(b)** SH. cN2O, surf-N2O, and obs-$N_2O$ has been rescaled to reflect that of a tropospheric abundance of 320 ppb. The hemispheric domains for STE is defined as 0°-90° while the surf-N2O and obs-$N_2O$ is from 30°-90° N/S. Note: the left y-axes limits are different between the tracers, but the interval scale is the same.

*4.2. QBO cycle*
The QBO composite of hemispheric mean cN2O STE flux from Figure 4 is compared with the
composite of surface abundances (surf-N2O and obs-$N_2O$) in Figure 6. The peak in cN2O flux is
broad and flat, but centers on +2 months for the NH and +4 months for the SH. Unlike the
annual cycle, the QBO cycle in STE flux is almost in phase in both hemispheres, with the NH
preceding the SH. This phasing of the QBO cycle in surface $N_2O$ was seen with the three
models in R2021. In both hemispheres, the modeled surf-N2O peaks before the rise in cN2O
and then decreases through most of the period with elevated cN2O flux as expected. The
amplitude of the QBO STE flux is smaller in the NH than SH by about half, and the amplitude of
surf-N2O is likewise smaller. The ratio of the amplitudes of surf-N2O to cN2O STE flux is
similar in both hemispheres (~ 0.4 ppb per Tg/yr), which is encouraging. This ratio is larger than
the corresponding one from the annual cycles (~ 0.1 ppb per Tg/yr) because the length of the
QBO cycle leads to longer accumulation of $N_2O$-depleted air from the cN2O flux.
In the SH, where the QBO cycle in cN2O flux has a large amplitude, the modeled surf-N2O
matches obs-$N_2O$ in amplitude and phase as reported in R2021. In the NH, the comparison of
surf-N2O with obs-N2O is not so good:  obs-N2O has a much smaller amplitude and a different
phase.  This QBO cycle pattern is similar, but reversed, to that of the annual cycle and can be
understood in the same way.  The NH QBO cycle has relatively small amplitude and thus the
interference with the large-amplitude annual cycle adds noise, obscuring the QBO cycle.  In the
SH it is the opposite, with its weak annual cycle, the SH QBO cycle is clear.  The modeled cN2O
fluxes enable us to understand the large-scale variability of the observations.
Thus, for both annual and QBO fluctuations, when the variation in STE flux is dominated by
either cycle, the surface variations are clearly seen and modeled for that cycle.  This further
supports the findings in R2021 and other studies, that hemispheric surface $N_2O$ variability is
driven by stratospheric loss on annual (NH) and QBO (SH) cycles, and it is clearly tied to the
STE flux.  Given the connection between $O_3$ and cN2O STE, this relational metric can be used to
constrain the $O_3$ STE for a model ensemble.

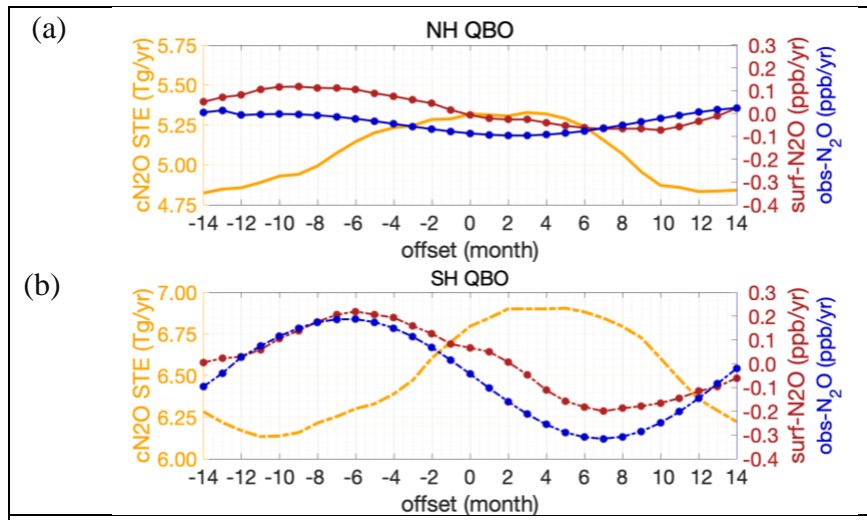

**Figure 6.  (a)** NH and **(b)** SH QBO composites of cN2O STE flux
(Tg/yr; orange lines, left axis, Fig. 4), and surf-N2O and obs-$N_2O$
(ppb; red and blue knotted lines, right axes, see Fig. 3 of R2021).
Results are shown for years 2001-2016 (6 QBO phase transitions), see
Fig. 4.  The surf-N2O data is from UCI CTM, and obs-$N_2O$ are taken
from NOAA ESRL, see text.

**5. Lowermost stratosphere**
If we accept that matching the observed annual and QBO cycles in surface $N_2O$ constrains the
modeled STE cN2O flux, then how can we use that to also constrain the modeled STE $O_3$ flux?
All evidence, theoretical, observational, and modeled, shows that the STE flux is simultaneous
for all species (e.g., Figure 1) and in proportion to their relative abundances (i.e., tracer:tracer
slopes) in the lowermost stratosphere, defined roughly as the region 100-200 hPa in each
hemisphere outside the tropics (Plumb and Ko, 1992).
*5.1. The $O_3$:$N_2O$ slopes and STE fluxes*

We can test the Plumb and Ko hypothesis in our model framework by comparing the relative
STE fluxes for $O_3$, cN2O and cF11 with the modeled tracer-tracer slopes in the lowermost
stratosphere. These slopes can then be tested using SCISAT-1 ACE-FTS (Scientific Satellite-1
Atmospheric Chemistry Experiment-Fourier Transform Spectrometer) measurements of $O_3$ and
$N_2O$ in the lowermost stratosphere to establish the ratio of the two STE fluxes. The ACE-FTS
$O_3$:$N_2O$ slopes were used for model transport and chemistry evaluation (Hegglin and Shepherd,
2007) and found to be very sensitive to satellite sampling, except in the lowermost stratosphere.
Figure 7ab shows the $N_2O$-$O_3$ slope in each hemisphere taken from the ACE climatology dataset
and the UCI CTM. The current ACE dataset (version 3.5) has been curated from measurements
made by ACE-FTS from February 2004 to February 2013 (Koo et al., 2017). The SCISAT orbit
results in irregular season-latitude coverage, and thus we average the lowermost stratosphere
data over a wide range of latitudes centered on the peak STE flux (20°-60° in both hemispheres).
For both ACE data and the CTM we keep to the lowermost stratosphere (200-100 hPa) and
average over the 4-month peak of STE flux, Feb-May in the NH and Sep-Dec in the SH (see
Figure 1). Extending into the upper tropical troposphere at 20° helps define the tropospheric
end-point of the slope (low $O_3$, high $N_2O$). Our method described here for deriving the slopes
from the ACE-FTS data is slightly different from that of Hegglin and Shepherd (2007; e.g., we
do not anchor the tropospheric point), and we have the advantage of a longer record.
Based on the long-term mean STE fluxes in the model, we would expect an $O_3$:$N_2O$ slope of
about -24 (ppb/ppb) in the NH and -17 in the SH. The slopes fitted to our modeled grid-cell
values of $O_3$ and N2O in the lowermost stratosphere are remarkably similar: -23.2 (NH) and -
17.5 (SH). The ACE data are more scattered but show similar, smaller slopes of -19.4 (NH) and
-15.3 (SH). Thus, the NH-SH asymmetry in $O_3$ versus $N_2O$ STE fluxes is clearly reflected in the
tracer-tracer slopes, both modeled and observed. Hegglin and Shepherd (2007) had already
identified these NH:SH differences when comparing their model to the ACE-FTS observations
(their Fig. 13cd), but implications for STE fluxes were not brought forward.
In the modeled SH (Figure 7b), one can see strings of points that are samples along neighboring
cells and reflect a linear mixing line between two different end points, one of which has
experienced extensive $O_3$ depletion (i.e., the Antarctic $O_3$ hole). We know that there is some
chemical loss of $O_3$ in the NH lowermost polar stratosphere during very cold winters (Manney et
al., 2011; Isaksen et al., 2012), but it is not extensive enough to systematically affect the $O_3$:$N_2O$
slope over the mid-latitude lowermost stratosphere in either the ACE observations or the CTM
simulations.

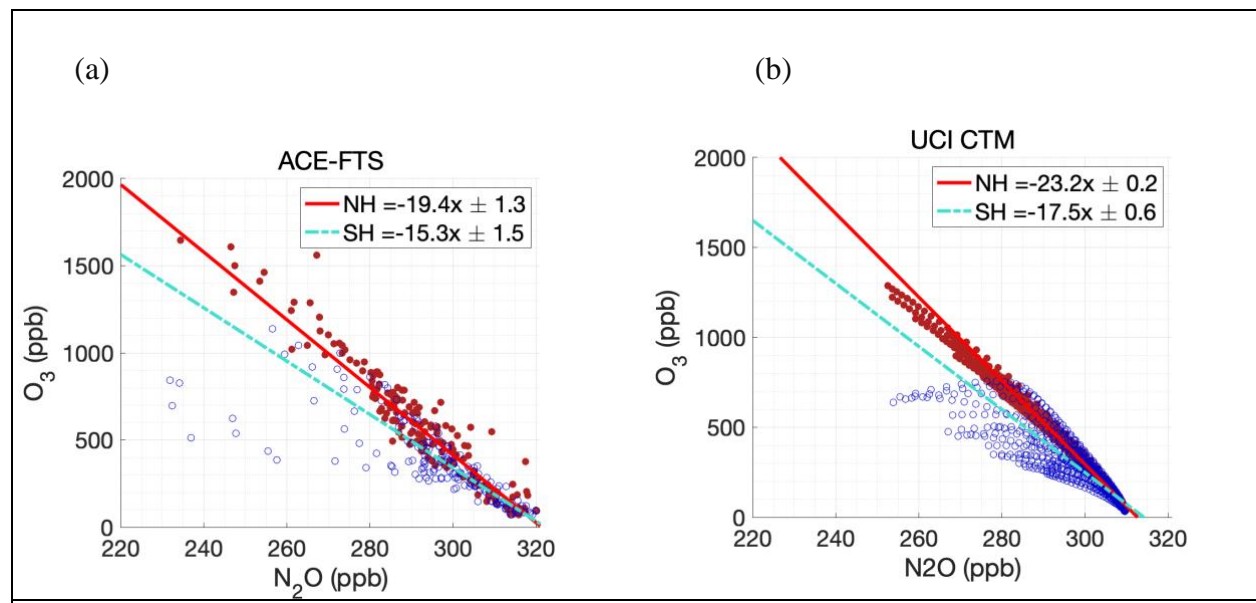

**Figure 7.** $O_3$ versus $N_2O$ (x-axis) scatter plots from **(a)** SCISAT ACE-FTS and **(b)** the UCI CTM. ACE-FTS data is from monthly climatologies for the period Feb 2004 to Feb 2013 restricted to 200-100 hPa, latitudes about 20°-60°, and months Feb-May (NH, red) or Sep-Dec (SH, blue). The linear-fit lines (ppb/ppb, values in legend) are restricted to larger $N_2O$ values (>280 ppb) to more accurately represent the STE fluxes, see Olsen et al. (2001).

*5.2. IAV of the Antarctic ozone hole and the SH STE $O_3$ flux*
The Antarctic ozone hole appears to be the source of the NH-SH asymmetry in the STE fluxes of
$O_3$ versus $N_2O$. It is known that the massive chemical depletion of $O_3$ inside the Antarctic vortex
between about 13 and 23 km altitude creates an air mass with lower $O_3$:$N_2O$ ratios than usually
found in the mid-latitude lowermost stratosphere. When the vortex breaks up, nominally in late
November, much of this $O_3$-depleted air can mix along isentropes into the mid-latitude
lowermost stratosphere, changing the $O_3$:$N_2O$ ratios and reducing the SH STE $O_3$ flux.
We have additional information on the SH $O_3$ STE flux from the year-to-year variations in the
size of the ozone hole. The best measure of the scale of Antarctic ozone depletion is the October
mean ozone column (DU) averaged from the pole to 63°S equivalent latitude (see Fig. 4-5 of
WMO, 2018). When we compare the CTM with the observations (Figure 8), we find remarkable
verisimilitude in the model: the root-mean-squared difference is 9 DU out of a standard
deviation of 29 DU and the correlation coefficient is 0.96. Thus, we have confidence that we are
simulating the correct IAV of the ozone hole. Next, we plot the modeled $O_3$ STE flux (summed
over the 12 months following the peak ozone hole, November-October) with the modeled
October ozone column and find a fairly linear relationship. If we estimate the STE $O_3$ flux
before the O₃ hole, when the mean October O₃ column was about 307 DU, then our O₃ flux
increases to 209 Tg/yr (see Figure 8, red marker), eliminating the hemispheric asymmetry in O₃
STE flux.
The annual deficit in SH STE O₃ flux brought on by the Antarctic ozone hole ranges from about
5 to 55 Tg/yr and with a central value of 30 Tg/yr or 14% of the total. Using the decadal trends
1965-2000 from Hegglin and Shepherd (2009), this deficit is 8%; and from Meul et al. (2018),
5%. Since both of these models calculate a much larger SH flux (~300 Tg/yr), we estimate their
absolute change in O₃ flux to be 24 and 15 Tg/yr, respectively. Because the ozone hole
effectively removes a fixed, rather than proportional, amount of ozone that presumably is
mapped onto the STE flux the following year, we believe the absolute change is the best
measure. Thus the three models estimate the ozone hole causes a deficit in the SH O₃ STE flux
in the range of 15-30 Tg/yr. The UCI CTM's ability to match the observed IAV of the ozone
hole, and to match that linearly with the deficit in STE flux provides support for the upper end of
the range. Note that the difference in O₃:N₂O slopes between NH and SH in Figure 7 is about 5.
If we attribute that solely to the ozone hole and split the flux of N₂O-depleted air evenly between
hemispheres, then the ozone-hole-driven O₃ STE flux difference is about 55 Tg/yr, about twice
that derived from the variability in our model. This difference in estimated flux indicates that
even without chlorine-driven ozone depletion, the O₃:N₂O slopes may be inherently different
simply because of the strong descent inside the wintertime Antarctic vortex. This can be readily
investigated with further model studies.
We looked for any relationship between ozone hole IAV and the STE fluxes of cN2O or cF11
and found mostly a scatter plot with no clear relationship. Given the analysis above, we expect
that much of the scatter is related to QBO cycles.

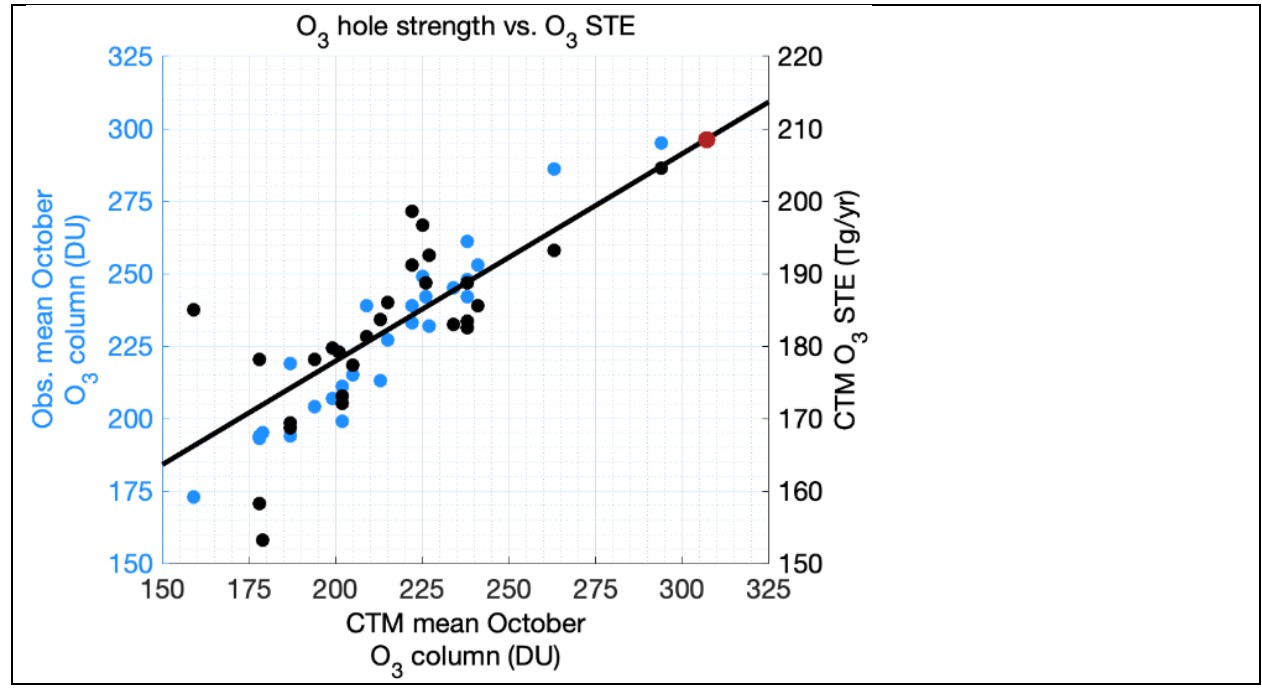

**Figure 8.** Interannual variability of the observed Antarctic ozone hole from 1990 to 2017 (blue dots; left y-axis) versus the CTM modeled ozone hole (x-axis); plus the CTM modeled SH STE $O_3$ flux (black dots; right y-axis) versus the modeled ozone hole (x-axis). The ozone hole is measured by the total ozone column (DU) averaged daily over October poleward of 63°S in equivalent latitude (see Figure 4.5 of WMO 2018). The SH STE $O_3$ flux (Tg/yr) is centered on May 1 of the following year (i.e., the 12 months following the nominal breakup of the ozone hole). The black line is a simple regression fit of the modeled STE to the modeled ozone hole (black dots), and the red dot is our estimate of pre-ozone-hole SH STE $O_3$ flux based on the observed 1979-82 $O_3$ column.

*5.3 Other model-measurement metrics related to STE*
What else might affect $O_3$ STE? Stratospheric column $O_3$ (DU) varies on annual and QBO
timescales. These changes in $O_3$ overhead can have a direct influence on $O_3$ transport to the
troposphere, but the link requires further analysis. Tang et al. (2021) showed the UCI CTM is
able to capture the observed annual cycle of stratospheric $O_3$ column as extracted from total
column using the Ziemke et al. (2019) method. QBO modulation of stratospheric column $O_3$ has
not been fully investigated since Tung and Yang (1994b). Yet, the fluctuations in mass over the
annual cycle are comparable to the corresponding variability in $O_3$ STE flux (1 DU = 10.9 Tg)
and likely connected (Figure 9).

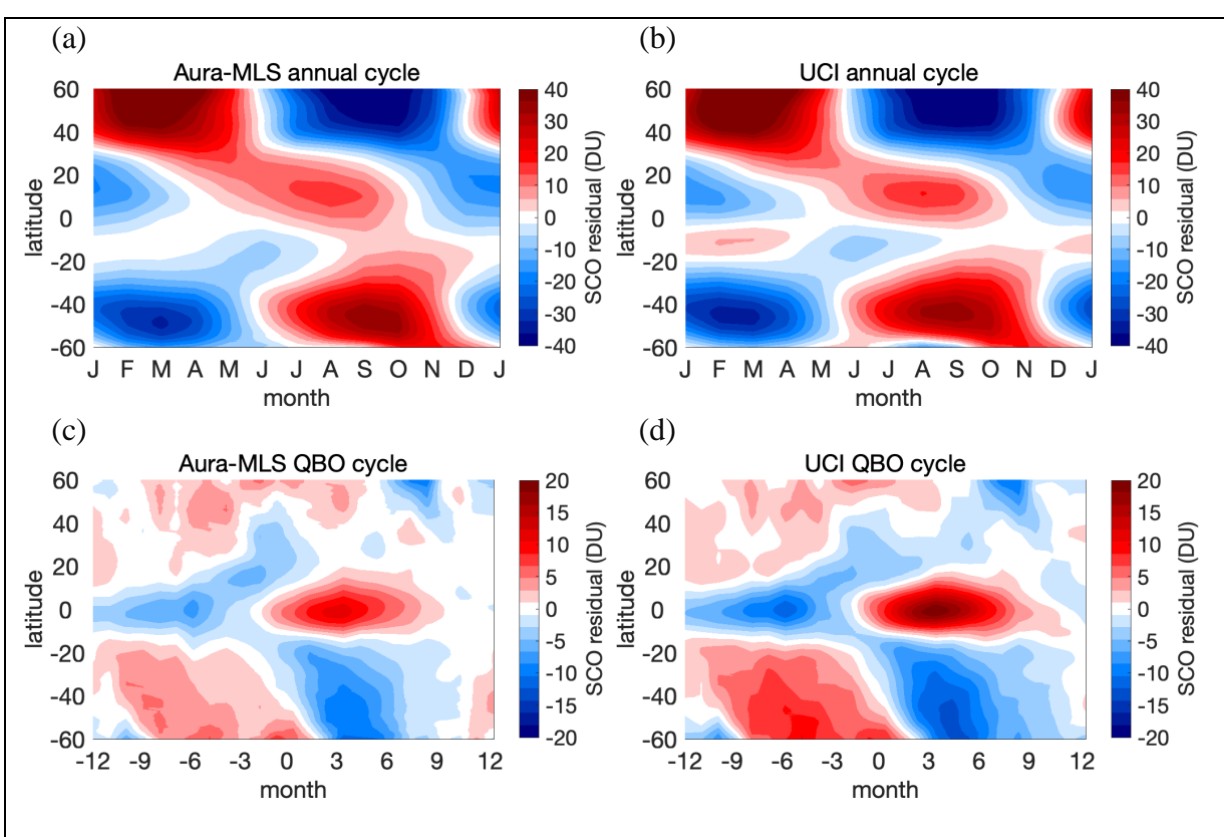

**Figure 9.** Stratospheric $O_3$ column residuals taken from Aura-MLS (**a, c**) and UCI CTM (**b, d**) for their mean annual cycle (**a, b**) and mean QBO cycle (**c, d**) during years 2005-2017. Residuals are defined at each latitude with a mean of zero DU.


**6. Conclusions**
This work examines how closely $O_3$ STE is linked to STE fluxes of other trace gases. By
including our complementary $N_2O$ and F11 tracers, we can follow stratospheric loss of these
gases along with stratospheric $O_3$ across the tropopause. The magnitudes of the fluxes are
proportional to their abundances in the lower stratosphere as expected (Plumb and Ko, 1992),
and their variability is highly correlated with one another, indicating that they are entering the
troposphere simultaneously. Even the distinct QBO pattern of STE fluxes is consistent across
$O_3$, $N_2O$ and F11. We further constrain the $N_2O$ transport pathway by linking STE of depleted-
$N_2O$ air with surface fluctuations of $N_2O$ abundance. The surface response in modeled $N_2O$
matches well with the observed surface variability in the SH, indicating that surface variability is
driven largely by STE flux.
*Consistency of STE $O_3$ flux.* As summarized here, there are a number of model diagnostics and
observational constraints that provide a reality check on the consistency of the modeled $O_3$ STE
flux. In Table 1, we examine these for our model and also for the CMAM model (Hegglin and
Shepherd, 2007, 2009) because it is one of the few with enough published results. For UCI we
calculate NH:SH fluxes of $O_3$ (208:182 Tg-$O_3$/yr) and $N_2O$ (5.1:6.4 Tg-N/yr). Thus the mole
fraction slopes in the lowermost stratosphere should be -23.8 (NH) and -16.6 (SH). Our model
$O_3$:$N_2O$ slopes are -23.2 (NH) and -17.5 (SH). Given the seasonal variability and scatter in the
correlation plots (Figure 7), we count this as consistent. For CMAM, the modeled $O_3$:$N_2O$
slopes, -23±2 (NH) and -18±3 (SH) are similar to ours and also to the ACE-FTS observations as
analyzed by Hegglin and Shepherd (2007), -22±4 (NH) and -14±3 (SH), or by us, -19 (NH) and -
15 (SH). CMAM does not report the implied STE $N_2O$ fluxes derived from their photochemical
loss of $N_2O$, but their model seems to match observations of $N_2O$ in the middle stratosphere, and
so we assume that the Aura-MLS derived $N_2O$ fluxes are a close estimate (12.9 Tg-N/yr). Note
we are using Aura-MLS $N_2O$ values here to calculate the photochemical loss, which occurs in.
the middle to upper stratosphere (see R2021 for methodology). Just using the CMAM global
numbers for $O_3$ STE flux, we calculate the $O_3$:$N_2O$ slope in the lowermost stratosphere should
average to -30. We conclude that their diagnosis of the STE $O_3$ flux, 655 Tg/yr, is inconsistent
with the circulation that generated the $O_3$:$N_2O$ slopes and is 50% too large. We do not view this
as a critical assessment of CMAM since it involves us combining diagnostics from two separate
publications and possibly different model simulations, but it is an example of how we might
expect future studies of the STE $O_3$ flux to self-evaluate.
*Uncertainty Quantification in STE $O_3$ flux*. Deriving a best estimate and uncertainty from this
work involves expert judgment. Changes in meteorological data used by the UCI CTM (IFS
Cycles 29r1, 36r1, and 38r1, all at 60-layer 1.1° resolution, see Table 1) give a standard
deviation in STE of 13% (only 3 values). If we use observations to derive a value as in Murphy
and Fahey (1994), we must expand our dimensions to the uncertainty in the NH:SH split of $N_2O$
flux to calculate each hemisphere's $O_3$ flux. The factors are: (1) total STE $N_2O$ flux is 12.9 Tg-
N/yr from the Aura-MLS data and we assign a ±10% one-sigma uncertainty; (2) the NH:SH split
of the $N_2O$ flux is 44:56 in our current model, was not diagnosed for previous ones, and so we
assume a value of 50:50 that ranges from 40:60 to 60:40; (3) analysis of the ACE-FTS
observations (ours and Hegglin and Shepherd, 2007) gives $O_3$:$N_2O$ slopes of about -21 (NH) and
-15 (SH) to which we assign a one-sigma uncertainty of ±3. Propagating these as root-mean
square errors, we find a ±15% uncertainty in the global value, $400 \pm 60$ Tg/yr. Uncertainty in the
hemispheric values is more difficult to assess, and from a range of model results shown in Table
1, we can only estimate that the NH:SH ratio is between 60:40 and 50:50, a range that bounds
our and CMAM results plus 2%. Note that this estimate is for current conditions with a regularly
occurring Antarctic ozone hole. We believe the low 50:50 ratio is plausible because we have
shown that our large SH STE $N_2O$ flux is consistent with the surface QBO variability in $N_2O$.
For pre-1980, and for when the ozone hole recovers later this century, we anticipate that the SH
$O_3$:$N_2O$ slope will revert to -18 to -21, and the total STE $O_3$ flux to 430-460 Tg/yr. This
simplistic estimate is based on a fixed atmospheric circulation.
A major surprise from our model is that the STE flux of $O_3$ is predominantly NH biased
currently, only because of the Antarctic ozone hole. Prior to 1980, and after 2060, it would/will
be symmetric between the hemispheres. Our model calculates slightly greater STE fluxes for
trace gases like $N_2O$ or F11 in the SH, which is counter to prevailing theory that the wave-driven
fluxes force relatively greater STE in the NH. This difference cannot be directly tested with
observations of trace gases, but a range of $N_2O$ hemispheric observations are well modeled and
support this premise. More extensive work with multi-model ensembles that include both
chemical and dynamical diagnostics in the stratosphere would be needed to overturn the
established theory. Our work reemphasizes the importance of trace-gas correlations in the
lowermost stratosphere as a key observational metric for climate models that may be able to
constrain total STE fluxes. The tracer slopes may go beyond just relative STE fluxes because we
have other measurements from the upper stratosphere to the surface that constrain, for example,
the absolute flux of $N_2O$ better than we first did using just the modeled lifetime.
In Table 2, we gather a set of observation-based model metrics that relate to STE fluxes and will
help the community build more robust models to better derive the STE flux of $O_3$.

| **Table 1.** Summary of key results for the STE flux of $O_3$ and $N_2O$ presented here (bold) | | | | |
|---|---|---|---|---|
| | NH | SH | Global | notes |
| STE $O_3$ flux (Tg$O_3$/yr) | **208** | **182** | **390** | IFS Cy38r1, yrs 1990-2017 (this paper; Ruiz et al., 2021) |
| | 239 | 198 | 437 | IFS Cy36r1, yrs 2000-2007 (Hsu & Prather, 2014) |
| | 301 | 233 | 534 | IFS Cy29r1, yrs 2000-2006 (Hsu & Prather, 2014) |
| | 383 | 272 | 655 | CMAM, yrs 1995-2005 (Hegglin & Shepherd, 2009) |
| | | | | |
| STE $N_2O$ flux (TgN/yr) | **5.1** | **6.4** | **11.5** | yrs 1990-2017, scaled to 320 ppb |
| | | | **12.9** | using Aura-MLS lifetime of 119 yr and 320 ppb |
| | | | | |
| LMS $O_3$:$N_2O$ slope* | **-23.2** | **-17.5** | | UCI model |
| | **-19.4** | **-15.3** | | ACE-FTS observations |
| | -23±2 | -18±3 | | CMAM model, Fig 13 of (Hegglin & Shepherd, 2007) |
| | -22±4 | -14±3 | | ACE-FTS observations, ibid |
| | | | -20.0 | (Murphy & Fahey, 1994) |
| | | | -22.0 | (McLinden et al., 2000) |
| | | | | |

| STE flux $O_3$:$N_2O$ (mole/mole) | **-23.8** | **-16.6** | | UCI model, calculated from entries above |
|---|---|---|---|---|
| | | | -29.6 | CMAM (Hegglin & Shepherd, 2009), using Aura-MLS $N_2O$ lifetime |
| Best Estimate STE $O_3$ flux (Tg/yr) | **60% to 50%** | **40% to 50%** | **400 ± 60** | current Antarctic ozone hole conditions, see text |

\* LMS = lowermost stratosphere only. For UCI model, months are selected for highest STE (FMAM in NH, SOND in SH, Fig. 1). For CMAM, the monthly ranges from their Fig. 13cd are estimated. Where no reference is given, the source is this paper.


**Table 2.** Metrics from Measurements or Constrained Values for CCMs related to Stratosphere-Troposphere Exchange

| Name | Metric | Measured values | Model requirements | Example figure |
|---|---|---|---|---|
| $N_2O$ loss | Annual and QBO cycles of global mean stratospheric $N_2O$ loss | Monthly $N_2O$ loss calculated from Aura-MLS profiles (2005-present) | Stratospheric chemistry for $N_2O$ as tracer; a QBO cycle; monthly mean diagnostics | Fig. 4 (Prather et al., 2015); Fig. 2 (Ruiz et al., 2021); Fig. 3 (this paper) |
| STE slopes | Matching $O_3$:$N_2O$ slopes in lowermost stratosphere | ACE FTS profiles (2004-2013) | Stratospheric $O_3$ and $N_2O$ calculation, possibly also CFCs; monthly snapshots | Fig. 7 (this paper) |
| Strat $O_3$ column | Annual and QBO composite cycles of stratospheric $O_3$ column | Monthly zonal mean stratospheric $O_3$ column from Ziemke et al., 2019 (2005-present) | Stratospheric $O_3$ chemistry; a QBO cycle; monthly mean diagnostics; separate strat & trop $O_3$ columns | Fig. 9 (this paper) |
| $N_2O$ loss at surface | Annual and QBO composite cycles of surface $N_2O$ solely from stratospheric loss | NOAA surface $N_2O$ observations | Stratospheric $N_2O$ chemistry; N2OX as a tracer; monthly mean diagnostics | Fig. 3 (Ruiz et al., 2021); Fig. 5 (this paper) |
| | | *Constrained (modeled) values* | | |
| STE flux of $O_3$ | | Monthly, latitude or hemispheric resolved, net $O_3$ flux | Run O3strat as a tracer; diagnose monthly flux into troposphere, at tropopause or through trop- loss of O3strat | Fig. 1 & 2 (this paper) |
| STE flux of $N_2O$ depleted air (also CFC-11) | | Monthly, latitude or hemispheric resolved, STE flux of $N_2O$ (CFC-11) | Run cN2O (cF11) as a tracer; diagnose monthly flux into troposphere | Fig. 1 & 2 (this paper); |
| SH $O_3$ hole and flux | | Change in SH $O_3$ STE flux with size of ozone hole; observed IAV of $O_3$ hole | IAV of ozone hole; daily total $O_3$ column (lat, long); monthly SH $O_3$ STE flux | Fig 7 (this paper) |

Notes: Constrained values are model-only derived quantities that can be diagnosed from CCMs or CTMs.


**Author Contributions:**
DJR and MJP designed and carried out the study and prepared the manuscript for publication.
**Competing interests:**
The authors declare that they have no conflict of interest.
**Acknowledgments:**

Research at UCI was supported by grants from the National Aeronautics and Space
Administration's Modeling, Analysis and Prediction Program (award NNX13AL12G), and
Atmospheric Chemistry Modeling and Analysis Program (80NSSC20K1237, NNX15AE35G),
and the National Science Foundation (NRT-1633631). We gratefully acknowledge the work of
the MLS team in producing the Level 3 data sets that enabled our MLS-related analyses. Work at
the Jet Propulsion Laboratory, California Institute of Technology, was performed under contract
with the National Aeronautics and Space Administration. We thank the ACE-FTS team for
making the climatology data used here available for our analyses. The Atmospheric Chemistry
Experiment (ACE), also known as SCISAT, is a Canadian-led mission mainly supported by the
Canadian Space Agency. We also acknowledge Ed Dlugokencky for providing the surface $N_2O$
data that was used here to produce an observation-based reference with which to compare our
simulated results. The data used to produce the figures and tables in this work are accessible via
the DRYAD repository with DOI https://doi.org/10.7280/D1JX0K

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
