# Peer review of "From the middle stratosphere to the surface, using nitrous oxide to constrain the stratosphere-troposphere exchange of ozone"

_Atmospheric Chemistry and Physics, 2021_

## Author Response (AR1)

We appreciate the reviewers taking the time for a task that is sometimes thankless—and for in turning in their reports in a timely manner so that we can proceed promptly with revising and publishing this work,

Daniel Ruiz & Michael Prather

Response to **Anonymous Referee #1**

> This study concerns developing observational constraints for the stratosphere-troposphere-exchange (STE) of ozone using N2O and CFC-11. The authors argue that the concentrations and distribution of the latter species are well constrained with satellite and surface measurement data and exhibit useful cross-tropopause concentration gradients and relationships with ozone. Using measurements and simulations, the authors determine STE fluxes of ozone, N2O and CFC-11 on seasonal and interannual timescales, as well as presenting how the fluxes influence the surface concentrations and how ozone fluxes in the southern hemisphere are related to the depth of Antarctic ozone hole. One particularly notable result is that their analysis points to a hemispheric symmetry of STE fluxes, in contrast to the expectation from the asymmetric strength in the Brewer-Dobson circulation.
>
> I feel that the results and analysis presented here will be a valuable addition to the community, particularly given recent pushes to understand tropospheric ozone processes and variability through IGAC's TOAR project. Overall, I have no major reservations about the methods and analyses but I do think that the manuscript would benefit from a major overhaul in terms of its structure and presentation: I, for one, found it very hard to follow all the different threads. I expand on this comment below along with some other comments and suggestions.
>
> **Major comment**
>
> Please consider revising the structure and presentation of the manuscript. As written, the tools, data sources and, crucially, major findings are not clear to this reader. For instance, I would encourage the authors to present the global and hemispheric ozone STE fluxes in the abstract and conclusions – these are going to the numbers that many will want to draw from this analysis, and (like it or not!) may not have time to read through the rest of the study. Additionally, highlighting the hemispheric fluxes would enable the authors to more obviously highlight the (pre/post ozone hole) NH and SH symmetry that they find, which will interest many in the stratospheric dynamics community.

We agree that the STE fluxes should be highlighted and clearly stated in both places. We include a new Table (1) summarizing these, and have added two new paragraphs in the last Section on consistency and uncertainties. For example, we discuss the self-consistency of our results for the $O_3$ and $N_2O$ fluxes with the $O_3$:$N_2O$ slopes in the lowermost stratosphere.

> I would also encourage the authors to consider the story that is told through the manuscript. I do not insist on a traditional structure, but I would certainly appreciate a clear distinction between data and methods and the rest of the results. As presented, the model gets described in the same section that discusses global STE results before we see new sections on interannual variability and the importance of the ozone hole, with the tracer data described later. I find all these different threads confusing and I lose how they are relevant to the bigger story about ozone STE that the title and abstract promise. There are many possible solutions to this, and I only encourage the authors to think about signposting the reader and telling a logical story.

We have pulled out the 'methods' material in a new section 2 and made it separate. We think this

does indeed help the flow. In terms of the order, we find it difficult re-order without a total rewrite, and we cannot see a clear path to such a rewrite. The material is indeed a circle of connected analyses. The expanded final section may help.

**Other comments**

1. Can uncertainty estimates for the STE fluxes be provided? Notwithstanding model and measurement uncertainty, is there something that can be estimated from the tracer-tracer correlations?

We should not shirk this task, and so we have developed a simple traceable best estimate, following along the lines of the original Murphy & Fahey approach. See new Table (1) and paragraphs in the final Section 6 on 'consistency' and 'uncertainty'. Thanks for the push on this.

2. Please review the clarity of the figures, particularly considering appropriate font sizes.

Yes, Figures 1-3 need larger fonts or to be presented as larger figures, the others look OK to us.

3. I encourage the authors to proofread the manuscript as there are several typographical errors.

Yes, we humbly agree and have been finding and fixing those.

**Specific comments**

P1, L15: Presumably CCMs as well as CTMs?

Yes. For the abstract we just used 'global chemistry models'.

P2, L61: Climate projections of what?

Yes, have changed that to: "projections of future warming"

P6, L18: "…well correlated ($r \sim 0.9$)…" (to be clear what the 0.9 is)

Correct, we now have defined cc as Pearson's correlation coefficient (old L166) and now use 'cc ~ 0.9' in the parentheticals, with other clarity improvements in the paragraph noted here (old L217-255)

P8, L260: Do you mean the anomalous QBO of 2015/6, rather than "08/2010"?

No. Based on Ruiz et al, 2021 the surface impact on $N_2O$ was highly anomalous during the 2010 QBO. We made this distinction clear in the revision.

P16, Table 1: Suggest this goes in a discussion section, and suggest full references are used so

This table (now Table 2) is located and discussed in the final section.  The abbreviated references have been made standard as requested.

Response to **Anonymous Referee #2**

This paper suggests a set of metrics based on observations to test stratosphere- troposphere exchange (STE) of ozone and other tracers in models, which is a relevant topic given the importance of STE for the tropospheric burden of these species and implications for climate radiative forcing. While there are some promising results in this paper, presentation of these should be improved before I can recommend this study for publication. See detailed comments below.

**Major comments**

**Methodology** is not explained in a way that readers can easily follow and should be introduced in a section separate from the results. At least for the model and observational details used. You reference surface observations for the first time in L278 only. Details of how to generate the metrics may remain in the main body of the manuscript. Particularly, the paragraph starting at L109 introduces a new and rather complex methodology that is not substantiated. I suggest adding some figures to illustrate how the gradients of these tracers and their **scaled** versions across the tropopause in the end look like. This is important because I cannot easily assess if some of the surprising results you find are due to this tracer **scaling** (see below comments). The e90 tropopause is not that widely applied and should be explained in further detail. How would your results change if using a different tropopause? This information again is important if you want your metrics be used by the wider modelling community who may not have the e90 tropopause implemented in their models. When calculating global mean STE do you apply a latitude-weighting? If so, please add to the methodology. This would allow the reader to better judge the realism of the approach.

We clearly have a different view than this reviewer on how papers should be written and what should be included.  This anonymous reviewer has strong views on publication, English language usage, referencing, and structure that are quite different from ours.  We can respect their viewpoint, and the journals can and should publish papers including a spectrum of such perspectives.  Nevertheless, this paper represents the perspective of the authors and meets the requirements for transparency and documentation.  We understand that the reviewer would like to have the paper re-written to satisfy their viewpoint, but they are not an author.  We have examined all of the comments and revised the paper in response to some of these, but many of the comments are either irrelevant or make no sense to the authors.

We take issue with this reviewer regarding the dumping of a large number of references that they claim must be cited.  Upon careful re-reading of these, in most cases, we cannot find the reason why.  We document this carefully below, but do not appreciate this unproductive work.  Does the reviewer have a different memory of these? They are great papers, but most seem to be irrelevant to the specific work presented here.  In a couple case we have found reason to include and cite the work.

*"When calculating global mean STE do you apply a latitude-weighting?"*  Is this a rhetorical question? We report STE in Tg/yr, so that clearly describes how we integrate over the globe.

The "mean" is either monthly or annually. This is pretty standard stuff and hardly needs a 'methodology' section. Moreover, the scaling of the color bar in Figure 1 (STE by latitude and month) states the units: "% of global, annual mean STE in each bin (1 month by ~1.1° latitude)", which clearly means area or cosine(latitude) weighted.

Starting with the *"Methodology"* issue. Considering suggestions from reviewer #1 and this comment from reviewer #2, we have added a new "Methods" section 2. That section notes that the only new method in this paper is the introduction of the complementary-flux tracers cN2O and CF11. They are fairly obvious and are simply explained. All the other methods that the reviewer wants re-explained have already been published in a suite of papers, particularly from the companion paper to this one that was published well ahead of the time of this review (Ruiz et al., 2021 JGR, First published: 08 March 2021), and extending back to our original work on stratospheric N2O and STE O3 fluxes:

*Ruiz, Daniel J., Michael J. Prather, Susan E. Strahan, Rona L. Thompson, Lucien Froidevaux, Stephen D. Steenrod (2021), How atmospheric chemistry and transport drive surface variability of N2O and CFC-11. J. Geophys. Res.: Atmospheres, 126, e2020JD033979. https://doi.org/10.1029/2020JD033979.*

*Prather, M.J., J. Hsu, N.M. DeLuca, C.H. Jackman, L.D. Oman, A.R. Douglass, E.L. Fleming, S.E. Strahan, S.D. Steenrod, O.A. Søvde, I.S.A. Isaksen, L. Froidevaux, and B. Funke (2015) Measuring and modeling the lifetime of nitrous oxide including its variability, J. Geophys. Res. Atmos., 120, 5693–5705. doi: 10.1002/2015JD023267.*

*Hsu, J.C., M.J. Prather (2014) Is the vertical residual velocity a good proxy for stratosphere-troposphere exchange of ozone? Geophys. Res. Lett., 41, doi:10.1029/2014GL061994*

*Tang, Q., M.J. Prather, J.C. Hsu (2011), Stratosphere-troposphere exchange ozone flux related to deep convection, Geophys. Res. Lett., 38: L03806, doi:10.1029/2010GL046039.*

*Tang, Q., M.J. Prather (2010), Correlating tropospheric column ozone with tropopause folds: the Aura-OMI satellite data, Atmos. Chem. Phys., 10, 9581-9688.*

*Hsu, J., M. J. Prather (2009), Stratospheric variability and tropospheric ozone, J. Geophys. Res., 114, D06102. [doi:10.1029/2008JD010942]*

*Hsu, J., M. J. Prather, and O. Wild (2005), Diagnosing the stratosphere-to-troposphere flux of ozone in a chemistry transport model, J. Geophys. Res., 110, D19305, doi:10.1029/2005JD006045.*

*McLinden, C., S. Olsen, B. Hannegan, O. Wild, M. Prather, and J. Sundet (2000) Stratospheric ozone in 3-D models: a simple chemistry and the cross-tropopause flux, J. Geophys. Res., 105, 14653-14665.*

*Avallone, L.M. and M.J. Prather (1997) Tracer-tracer correlations: three-dimensional model simulations and comparisons to observations, J. Geophys. Res., 102, 19233-19246.*

We do not believe it is appropriate to re-publish such material when it is readily available.

*"Scaling".* The reviewer clearly has a problem with the word 'scaling' and seems to associate it with an arbitrary altering of the data to make the results look better. We cannot understand their obvious misreading of the current text and the avoidance of perusing the precursor paper Ruiz et al., 2021. The text is clear here that the scaling is used to make an e-folding tracer stationary and tied to a specific tropospheric abundance so the budgetary terms (total loss, STE fluxes, etc) can be compared. The text reads:

"The multi-decade (F11X) to century (N2OX) decays are easily rescaled on a month-by-month basis (using a 12-month smoothing filter) to give stationary results and a tropospheric mean abundance of 320 ppb.

*Line 117 insert:*
> "The c-tracers and their STE fluxes are rescaled as their corresponding X-tracers to give them a stationary **[time series corresponding to a]** tropospheric abundance of 320 ppb **[for their X tracers].**

We have added few phrases as [**bold**] to be overly clear on what the scaling does.

Scale/Scaling is also used in Figure captions, and that usage also seems very clear to us.
> **"Figure 3. ….** The scales for cN2O and cF11 are kept in a 1:2 ratio.
> **"Figure 5…** cN2O, surf-N2O, and obs-N2O has been rescaled to reflect that of a tropospheric abundance of 320 ppb

Other uses include the tracer:tracer slopes as a scale for the fluxes (this is obvious and well established):
> "The observed tracer correlations between N2O and O3 in the lowermost stratosphere provide a seasonal, hemispheric scaling of the N2O flux to that of O3

And a subjunctive/hypothetical use is to estimate the O3 variability if it scaled with the STE flux. It does not because as explained the O3 lifetime is shorter than a season:
> "A mole-fraction scaling of the STE fluxes gives an O3:N2O ratio of ~25, and thus scaling the surf-N2O amplitude gives a large O3 surface seasonality of ~18 ppb.

The use of the ***synthetic tracer e90*** to separate stratospheric from tropospheric air clearly has great skill in following the interhemispheric and seasonal variations in tropopause O3.  A number of global models use e90 and we do not need to re-justify its use here nor re-explain it.
> *Prather, M.J., X. Zhu, Q. Tang, J. Hsu, J.L. Neu (2011), An atmospheric chemist in search of the tropopause, J. Geophys. Res., 116: D04306, doi:10.1029/2010JD014939.*

The e90 paper has over 50 citations from independent modeling/analysis groups, including a major review paper:
> *Baldwin, M.P,, T. Birner, G. Brasseur, et al., (2019) , 100 Years of Progress in Understanding the Stratosphere and  Mesopshere, AMS monographs, Vol 59, Ch.27, doi:10.1175/AMSMONOGRAPHS-D-19-0003.1*

Further, the 2014 paper (Hsu et al.) thoroughly compared the various dynamical approaches for diagnosing STE O3 fluxes, comparing e90 with many. It is also the reference for our statement that ozone chemistry below 16 km is not major (i.e., the STE fluxes across the e90, 120 ppb O3 and 250 ppb O3 are not that different, except for the known tropical O3 production between 120 and 250 ppb).  This paper also diagnoses all the ozone budgets terms for the lowermost stratosphere (all in their Fig.1). We now include that reference in the new, brief methodology section to justify the 16 km statement.
> *Hsu, J.C., M.J. Prather (2014) Is the vertical residual velocity a good proxy for stratosphere-troposphere exchange of ozone?  Geophys. Res. Lett., 41, doi:10.1029/2014GL061994*

> The results should also be better placed into context of the already published literature. Discussion of whether your results confirm previous estimates would strengthen your results and provide support for your methodology and the suggested metrics table. See detailed comments below.

We have some comparisons with other results (e.g., the critical NH:SH ratio, ~L149) and clearly quantify our results.  We also noted early on that the comparison with other models is so wide as to not really constrain most of the models, including ours (L42, the TOAR assessment, Young et al., 2018).  Nevertheless, in response to the reviewer's request, we have expanded some comparisons as noted below:

*Line 134 insert:*
… uncertainty). This value is well within the uncertainty in the observation-based estimates (Murphy and Fahey, 1994; Olsen et al., 2001), and far from the extreme ranges of the 34 models in the latest Tropospheric Ozone Assessment Report (Young et al., 2018), 150 to 940 Tg/y. The global STE flux of cN2O is ….

*Line 150:*
Hsu and Prather, 2009; Yang et al., 2016), although some have higher ratios like 58:42 (Hegglin and Shepherd, 2009; Meul et al., 2018)

*Line 458. [We shall insert the following paragraph in line 458, which compares our ozone hole deficit with two other results using chemistry-climate models. These two papers (newly referenced) did not clearly state an estimate of what the ozone hole was doing to STE fluxes, rather they just fitted a trend line, which we used to estimate the pre-hole minus hole (2000s) difference. ]*

The annual deficit in SH STE $O_3$ flux brought on by the Antarctic ozone hole ranges from about 5 to 55 Tg/yr and with a central value of 30 Tg/yr or 14% of the total. Using the decadal trends 1965-2000 from Hegglin and Shepherd (2009), this deficit is 8%; and from Meul et al. (2018), 5%. Since both of these models calculate a much larger SH flux (~300 Tg/yr), we estimate their absolute change in $O_3$ flux to be 24 and 15 Tg/yr, respectively. Because the ozone hole effectively removes a fixed, rather than proportional, amount of ozone that presumably is mapped onto the STE flux the following year, we believe the absolute change is the best measure. Thus the three models estimate the ozone hole causes a deficit in the SH $O_3$ STE flux in the range of 15-30 Tg/yr. The UCI CTM's ability to match the observed IAV of the ozone hole, and to match that linearly with the deficit in STE flux provides support for the upper end of the range.

**Minor comments**

Abstract: The abstract should be improved (I guess mostly a language issue) and state clearer and self-containing results that don't require reading the whole manuscript to understand. For example, *'The STE flux of O3, however, is predominantly northern hemispheric, but observational constraints show that this is only caused by the Antarctic ozone hole.'* I know what you mean but it's not written with a clear logic, the Antarctic ozone hole doesn't affect the NH STE flux directly, just its relative magnitude. Also you say *"we show that metrics founded on observations can **better** constrain the STE O3 …."* better than what?

We do not understand the reviewer's problem with the first sentence. It reads clearly to us after several re-reads, and this is an abstract. If we have room to add words to the abstract, we will add the clause **"reducing southern hemispheric O₃ but not N₂O STE"** to the end of the sentence.

Reviewer is correct, we can drop "**better**" and the sentence reads fine.

L35 To call stratospheric ozone influx is driving climate change and surface air pollution is overstated, since certainly fossil fuel emissions are the main cause for these. Statement should be weakened. Some references added to provide justification.

The sentence is correct as written.  It does not say or imply that the STE flux is **the** driving force, but a driving force.  We thank the reviewer for reminding us of the **Zeng et al (2010) paper** that indeed shows that changing STE flux over the 21[st] century has a large impact on tropospheric ozone (climate and air pollution), and we add that reference to justify the statement.  Likewise, the **Williams et al. 2019** paper.  We also include *Hess, Kinnison, and Tang (2015 ACP: Ensemble simulations of the role of the stratosphere in the attribution of northern extratropical tropospheric ozone variability*) with these 2 to support the statement.

From the Hess, Kinnison, Tang paper:
> *"... a large portion of the measured change [in $O_3$] is not due to changes in emissions, but can be traced to changes in large-scale modes of ozone variability. This emphasizes the difficulty in the attribution of ozone changes, and the importance of natural variability in understanding the trends and variability of ozone. We find little relation between the El Niño–Southern Oscillation (ENSO) index and large-scale tropospheric ozone variability over the long-term record."*
> *"While diagnostics of the STE of ozone across the tropopause would be preferable, they could not be estimated precisely from the monthly averaged model output fields saved from these simulations."*

***Insert Line 35:***
… pollution (Zeng et al. 2010; Hess et al., 2015; Williams et al., 2019).

L36-37 Add citations to Lelieveld and Dentener (2000) and Lamarque and Hess (1999). Also, what is a regular seasonal cycle? I would argue that regularity is not maintained at a longer timescales where ozone depletion has affected the N2O-O3 relationship.

The reviewer has led us to re-read these classic papers that were state of the art two decades ago.

***Lelieveld and Dentener (2000 JGR: What controls tropospheric ozone?***) is as the title implies a study of tropospheric ozone and not STE flux.  The model was state of the art in the 1990s, but has some serious problems in calculating STE since it has only 19 levels and a top at 10 hpa, precluding any realistic Brewer-Dobson circulation.  The STE reported, 565 Tg/y, is not particularly notable, it falls in the range of models at the time, see summary table in 2001 IPCC report.  There is no resolution of the STE O3 flux (latitude, seasons, …) reported.

The ***Lamarque, Hess, and Tie paper*** (*1999 JGR: 3D model study of the influence of stratosphere-troposphere exchange and its distribution on tropospheric chemistry*) admits that their model (monthly mean winds + diffusion) simply cannot do STE, and cannot model tropopause folds.  They must turn off strat-to-trop transport in the model and parameterized STE as an added source of O3 and HNO3 (very clever modeling here).  The STE impact on tropospheric O3 (again, not the topic of this paper) is totally parametric.

The major interest of these classics in on what controls tropospheric ozone, not on the accurate calculation the ozone STE flux.  Thus, there is no reason to cite them in this paper.

L45-46 Statement needs to be backed up with references of how N2O-O3 relationships were

used for model-observation metrics.
The statement here is specific to our knowledge of the N2O STE.  The reference here should be the papers where N2O observations are used to calculate the loss of N2O and hence it's STE flux:  e.g., Prather et al. 2015; Ruiz et al. 2021.

We have modified that paragraph as requested:
*Line 38:*
  … ratios, in particular $N_2O:O_3$ ratio in the lower stratosphere (Murphy & Fahey, 1994; McLinden et al., 2000), or dynamical …

*Line 46:*
  … well (Prather et al. 2015; Ruiz et al. 2021

> L92-93 This sounds exaggerated to say *'this method is extremely robust'* without providing the basis for the statement. Also, I would like to see a more critical discussion and some added caveats of the method used. After all, a CTM is likely not a sufficiently sophisticated tool to be used for the investigation of the seasonal cycles at the surface, given that N2O has strong sources from soils that show large geographical and seasonal variations (see Butterbach-Bahl et al 2013).

This comment makes little sense, the section is about the STE flux, not the surface. For the surface values (not relevant in this paper but demonstrated in Ruiz 2021), we show that 3 independent CTMs can reproduce a similar surface signal in N2O that is driven by the stratospheric STE flux.  That is "robust".  Obviously the surface signal is also affected by emissions, but that is not what we are studying here.
The calculation of STE flux using our method is "exact" as it measures the effective transport between strat and trop, including any numerical or diffusive transport.   Thus, in response to the comment, we have corrected this sentence:

*Line 92:*
  This method is precise and geographically accurate for $O_3$ and is self-consistent with a CTM's tracer-transport calculation (…

> L129-130 Please provide references supporting the statement, or was this meant to be describe your methodology? Again, the methodology should be better separated from your results.

The references to the CTM are all over this paper, but to satisfy the reviewer we add a recent reference to Linoz v3 in *Line 130* (Hsu and Prather, 2010).  The "methodology" is old and has been published and need not be 'separated from results'.

> L133-135 Please put your results into context with previous literature. The ozone estimate seems at the lower end of the range indicated by earlier observations studies (Murphy and Fahey 1994; Gettelman et al 1997) and modelling studies (Young et al., 2013; Stevenson et al 2006; Hegglin and Shepherd 2009; Kawase et al 2011)

We have already referenced the old semi-observational estimates of Murphy & Fahey, McLinden, Gettelman.  These are old and have large enough uncertainties (except Gettelman for some reason seems very narrow range given the inherent uncertainty in applying that method). We have however updated this section with a more modern reference than those suggested above.

***Line 134 insert (total O3 STE):***
… annual means).  This value is well within the uncertainty in the observation-based estimates (Murphy and Fahey, 1994; Olsen et al., 2001), and far from the extreme ranges of the 34 models in the latest Tropospheric Ozone Assessment Report (Young et al., 2018), 150 to 940 Tg/y. The global STE flux of cN2O is….

***Line 150:***
Hsu and Prather, 2009; Yang et al., 2016), although some have higher ratios like 58:42 (Hegglin and Shepherd, 2009; Meul et al., 2018)

> L137 this explanation is not clear to me, that is why the budget of cF11 is about twice as large as that of cN2O. If it is destroyed faster, shouldn't it be smaller not larger? Or since it is a budget that balances loss and sources, it may be illogical to make this comparison to begin with since it is determined by the realism of the tracers in your model?

We do not understand why the reviewer continues to disparage our CTM; there are numerous publications demonstrating good skill in matching observations.  While not perfect, the model is certainly as good as many of the references that the reviewer would have us cite.

There is clearly little chance that we can persuade this reviewer that our work here is new and worthwhile, but let us see if we can help persuade them about the cF11 flux.  The tracers F11 and N2O here are modeled as similar in abundance and molecular weight (as stated).  F11 has photochemical loss that extends down to 20 km or below in the tropics and hence has a lifetime of about 55 years.  N2O has loss only in the middle stratosphere and a lifetime of about 110 years.  The lifetimes are defined as Burden(kg)/Loss(kg/y).  Thus the loss of F11 is 2x that of N2O.  Thus the production of cF11 is 2x that of cN2O, and that loss must on average be transported into the troposphere as the STE flux.

> L140-144 Please put your results into context with previous literature. Sprenger and Wernli 2003 and Skerlak et al 2014 have illustrated that the subtropics are the main places of where stratosphere-to-troposphere transport happens and need to be referred to as well.

The Skerlak (2014) and Sprenger (2003) papers present an excellent application of Lagrangian trajectories to estimate the stratosphere-to-troposphere (call STT) and the reverse trop-to-strat (TST) fluxes of mass and O3.  One of their primary interests (not important for this paper) is how deep the stratospheric intrusions reach: "*There are clear hotspots of deep STT fluxes into the continental PBL.* "

A major problem in comparing these Lagrangian papers with global chemistry models is that all of our best models of STE flux (e.g., following the net change in tropospheric O3 after each advective-convective-diffusive time step (Hsu, 2005, 2009), tracking an O3strat tracer into the troposphere, or deducing the flux from an annual-mean tropospheric O3 budget (Young et al., 2013;2018)) evaluate the important component, the net STE, which equals their STT minus TST.  Their net O3 flux in the paper (their Fig S10) is nearly zero, like the net mass flux (which must be zero).  We quote:

> *"We find maxima in June (NH, 1.64 Tgmonth−1) and July (SH, 1.58 Tgmonth−1) and minima in September (NH, −2.27 Tgmonth−1) and February (SH, −2.26 Tgmonth−1).* ***Our results disagree with the seasonal cycles found in the Eulerian studies both in amplitude and timing.*** *The smaller values in our study are expected because method intercomparison studies have shown that* ***Eulerian models are too diffusive and are***

> *likely to overestimate STE fluxes, but it is not clear at first sight why we obtain net upward ozone fluxes in certain months. This points to a problem in the calculation of the TST ozone flux."*

The Eulerian models are not all "too diffusive" (although earlier versions were) since they calculate reasonable absolute STE O3 fluxes when compared with the observed N2O-O3 tracer correlations (Murphy and Fahey, 1994;McLinden et al., 2000).

In Skerlak (2014), the authors carefully acknowledge the problems and assumptions in the technique:

> The O3 STT depends on the tropopause ozone field assimilated in ERA-Interim, with problems caused by changing satellites;
> The mass and ozone fluxes across the tropopause depend on the parameterized minimum residence time.

This paper is valuable, especially when looking for deep stratospheric intrusions impacting surface air quality, but is clearly not a key reference paper for absolute values of O3 net STE fluxes.

Regarding the reviewer's claim that Skerlak showed the STE O3 flux to be primarily in the sub-tropics (like our results), this is simply not correct as seen in their Figure 16: peak STT lies well poleward of 30 degrees latitude.

> L146 There is no causal link between the small tropical fluxes to the interhemispheric asymmetry of the NH and SH, which this sentence seems to imply. Improve language.

We cannot understand the reviewer's problem here; their comment makes no sense. Our sentence is simple and correct and needs no 'improvement'. It merely states that the NH:SH split in STE is easy to diagnose and does not depend on precise definition of the atmospheric equator as long as it lies in the tropics, where STE O3 flux is small.

> L189-191 Why would O3 photochemical destruction only reach down to 16 km?

This is answered in part above, and is general knowledge in the community. The 16 km upper boundary of the LMS is used for dynamical reasons, but the LMS is also found to be a region of very slow ozone chemistry (Gettelman 1997; Hegglin and Shepherd, 2007).

> L235-137 Please improve language.

*Replace L235:*

Model results here indicate that in the NH, the IAV across O3, cN2O, and cF11 STE fluxes are synchronized, and thus the air masses entering the lowermost stratosphere have the same chemical mixtures from year to year.

*Replace L237:*

We know that cold-temperature activation of halogen-driven O3 depletion in the Arctic winter at altitudes above 400 K (potential temperature) can produce large IAV in column ozone (Manney et al., 2020), but…

L239-240 This is not correct. There is ample evidence from aircraft observations that polar vortex air mass processing and mixing into the LMS after polar vortex break-up is observed in trace gas distributions also in the NH.

We believe this statement is correct. The breakup of the Arctic vortex does mix $O_3$-depleted layers (>400K) into the lower stratosphere (LS) but not directly into the lowermost stratosphere (LMS, <380K). The Arctic loss occurs at higher altitudes than in the Antarctic.

L241-242 This is another limitation of the methodology applied that should be mentioned already in the methods section.

Sorry, this method is well known and published. The limitations for the Arctic are noted here.

L252-254 Don't you build in this tight correlation through your **scaling** of the tracers? Note your methodology in how you scale the two tracers was not 100% clear to me so I hope you could improve on the description in the methods section.

These plots have not been "rescaled" to get a better correlation. The only scaling here is to plot with different Y axes.

L258-259 It would be good to specify here that this is only true for these long-lived trace gases.

We do not understand this comment. No, it would not "be good" to add specious qualifiers here. The statement is clear and unambiguous, we are not talking about all the gases in the atmosphere.

L341-350 I really struggle with the logic here. Soil emissions do have a seasonality and thus I cannot see how your model results want to imply that there is no seasonality.

The English is quite clear: if the stratospheric signal has the same seasonality as observed then maybe the seasonality of emissions (which is agreed upon) is not manifest in the surface abundances. We can expand on the sentence.

***Line 343:***
…signal has no seasonality, although we know that some emissions are seasonal (Butterbach-Bahl et al., 2013).

Again, I am suspicious that the **scaling** you apply to your tracers is responsible for this result rather than this to reflect what happens in the real atmosphere.

The reviewer may have their suspicions and attribute it to some figment of scaling, but our results are straightforward, if unpalatable. Please re-read the paper to understand how "scaling" is used.

L352-361 These results seem to be in contradiction with Lamarque and Hess 1999 and also a recent study by Williams et al. 2019, who used a stratospheric ozone tracer to investigate STE impact on surface ozone seasonality. Please compare your results to these studies and provide a discussion of where the differences may come from.

This section is clearly and fairly written. It is a very rough back-of-the-envelope estimate (our only "scaling" here). The fact that it disagrees with the Lamarque and Hess and Tie 1999 paper is not consequential. As explained above, their modeling of STE is highly parameterized and calculated with a very coarse and early model. The Williams 2019 study is only 2 models and their diagnostics are not detailed enough to accept or refute our crude estimate. They plot the O3F (= fraction of trop $O_3$ that is stratospheric in origin), but focus on seasonal means and not monthly variability. Their one figure 7 of O3F at 850 hPa (not surface) does show a monthly

variability that is not like our suggested one (remember we are not modeling the perturbations to tropospheric O₃). It is, however, only one model doing a very different simulation. Certainly not worth a major comparison or discussion.

> L373-376 Do you not contradict yourself here with your earlier result that interannual variability driven mostly by the QBO leads to surprisingly similar amplitudes in the NH and SH?

We do not find a contradiction here.

> L400-435 Hegglin and Shepherd (2007) have used ACE-FTS O3/N2O correlations in an extensive comparison to a CCM, so should be cited here. This study reveals how sampling issues can affect the interpretation of tracer-tracer correlations using ozone. In particular, the ACE-FTS instrument exhibits a strong sampling bias with unequal sampling of seasons and hemispheres. Undersampling the full correlation space (since your monthly ACE-FTS data will not have sampled all latitudes in your considered latitude range evenly) is likely to impact your results. The differences (or even apparent agreement!) in the slopes between observations and your model may thus be at least partially explained by this sampling bias. A discussion of the limitation of your approach should thus be added.

We were clearly in error by failing to reference the earlier work on N2O:O3 slopes from ACE-FTS by Hegglin and Shepherd (2007). Nevertheless, the reviewer clearly does not remember this paper very well. Hegglin (2007) is fundamentally a paper about the lower and middle stratospheric circulation in their CMAM model using ACE FTS data of N2O and O3 to test it. They are primarily concerned with the intra-stratospheric circulation, and when they present results for the lowermost stratosphere, they never mention the stratosphere-troposphere exchange flux nor relate the O3:N2O slopes to the fluxes.

Further, the reviewer is totally mistaken in thinking that ACE profiling cannot deliver robust tracer:tracer slopes in the lowermost stratosphere (LMS) as they are used here. I include some direct quotes from the Hegglin paper that support the very common view that irregular sampling by ACE, ATMOS, aircraft, or balloons is adequate for this work, but of course problematic for the middle stratosphere.

> *"By subsampling the CMAM data, the representativeness of the ACE data is evaluated. **In the middle stratosphere, where the correlations are not compact** and therefore mainly reflect the data sampling, joint probability density functions provide a detailed picture of key aspects of transport and mixing,…"*

> *"Sufficiently long lived species exhibit compact correlations [Plumb and Ko, 1992], which eliminate day-to-day variations, providing an ''instant climatology,'' and **mean that even limited measurements** can provide a robust constraint on models…"*

> *"In section 4, we focus on the lower stratosphere, where the O3-N2O correlations are generally compact. **The good spatial and temporal coverage of the ACE data**, "*

> *"Comparing full to subsampled CMAM results shows that where the midlatitude and polar slopes are the same **the slopes are very well estimated by the subsampling (e.g., NH LMS)**. This is no surprise, since in this case the extratropics are well mixed and the correlations compact."*

We are pleased to recognize this work as identifying the NH-SH differences that are reassessed

in terms of STE fluxes here.

**Revise Line 427:**
Hegglin and Shepherd (2007) had already identified these NH:SH differences when comparing their model to the ACE-FTS observations (their Fig. 13cd), but implications for STE fluxes were not brought forward.

> L441-445 This is a well-known feature and existing references should be added and discussed to highlight this (Lamarque and Hess 1999; Williams et al. 2019; Hegglin and Shepherd 2009)

The opening sentence is new (as noted for L493 below). The fact that the ozone hole reduces SH O3 STE fluxes has been updated in the text above with references (Meul, Hegglin & Shepherd). There are not that many studies that diagnoses NH:SH STE and also do pre-ozone hole. The Williams 2019 study does not diagnose such changes in STE. The Lamarque & Hess paper is inappropriate here. We have tried to clarify the established facts. We do not have to reference everything.

**Revise Lines 442-445:**
It is known that the massive chemical depletion of O$_3$ inside the Antarctic vortex between about 13 and 23 km altitude creates an air mass with lower O$_3$:N$_2$O ratios than usually found in the mid-latitude lowermost stratosphere. When the vortex breaks up, nominally in late November, much of this O$_3$-depleted air can mix along isentropes into the mid-latitude lowermost stratosphere, changing the O$_3$:N$_2$O ratios and reducing the SH STE O$_3$ flux.

> L468-475 This evaluation does not provide much depth since it doesn't clearly link total column ozone to STE (and to provide metrics for STE was the main goal of your paper I thought). Your statement ' is clearly somehow connected' expresses this weakness. I would like to see how it is connected otherwise I suggest to remove this section from the paper. You may introduce this differently, that is start with this evaluation since a model must represent the right ozone distribution to get STE ozone right?

We believe that it is relevant to relate the annual and interannual changes in column O$_3$ to the STE fluxes. The scales are similar (multiply the DU scales by 10.9 to get Tg). This section is admittedly speculative, but it notes that the scales of variability are comparable. One does not cause the other, but both column and STE are measures of the stratosphere circulation and STE. We see no need to remove this section, it is informative.

**Replace Line 472-474:**
QBO modulation of stratospheric column O$_3$ has not been fully investigated since Tung and Yang (1994b). Yet, the fluctuations in mass over the annual cycle are comparable to the corresponding variability in O$_3$ STE flux (1 DU = 10.9 Tg) and likely connected (Figure 9).

> L493-494 This is not a result of your study and should be attributed to literature that have discussed this (Meul et al. 2018; Hegglin and Shepherd 2009; Zeng et al. 2010)

Sorry, ***this is a novel result of our study***. The other studies have NH:SH STE O3 ratios >1 for all conditions (pre-ozone hole and present). We agree that **at present** NH:SH >1 and that is noted above to agree with other studies. What we find is that pre-ozone hole the NH:SH ~1. We believe the likelihood of the model being correct is supported by our ability to simulate surface N$_2$O in the SH and the IAV of the ozone hole. Our proposal is unique and unusual, and it contradicts the major premise of wave-driven STE being primarily NH.

L507 This table should include earlier references who suggest the same metrics as tests for model STE.

We do not think that adding citations here will help. Further, for us to attempt to define the provenance of all the possible model and measurement metrics would have us go back to the 1993 Models and Measurements workshop or before. This is not a review paper. We agree that changing the sentence that introduces Table 1 would be good.

*Line 507:*

In Table 1, we gather a set of observation-based model metrics ….

**Additional references:**
Butterbach-Bahl, K., Baggs, E.M., Dannenmann, M., Kiese, R. and Zechmeister- Boltenstern, S., 2013. Nitrous oxide emissions from soils: how well do we understand the processes and their controls?. Philosophical Transactions of the Royal Society B: Biological Sciences, 368(1621), p.20130122.

Butterbach-Bahl (2013) is an excellent paper on the biogenics of nitrogen in soils (e.g., nitrification, denitrification) and the release of $N_2O$. It is a core reference on the recent Nature review paper on $N_2O$ (Tian et al., 2020). Nevertheless, this paper and its companion paper (Ruiz et al., 2021) do not attempt a complete simulation of all $N_2O$ sources, but only that from the stratosphere. For example, we do not cover and cite the oceanic sources.

Hegglin, M. I. and Shepherd, T. G., 2009, Large climate-induced changes in ultraviolet index and stratosphere-to-troposphere ozone flux, Nat. Geosci., 2, 687–691, https://doi.org/10.1038/NGEO604.

Hegglin, M.I. and Shepherd, T.G., 2007. O3― N2O correlations from the Atmospheric Chemistry Experiment: Revisiting a diagnostic of transport and chemistry in the stratosphere. Journal of Geophysical Research: Atmospheres, 112(D19).

Kawase, H., Nagashima, T., Sudo, K. and Nozawa, T., 2011. Future changes in tropospheric ozone under Representative Concentration Pathways (RCPs). Geophysical Research Letters, 38(5).

Lelieveld, J. and Dentener, F. J., 2000, What controls tropospheric ozone?, J. Geophys. Res.- Atmos., 105, https://doi.org/10.1029/1999JD901011.

Meul, S., Langematz, U., Kröger, P., Oberländer-Hayn, S. and Jöckel, P., 2018. Future changes in the stratosphere-to-troposphere ozone mass flux and the contribution from climate change and ozone recovery. Atmospheric Chemistry and Physics, 18(10), pp.7721-7738.

Škerlak, B., Sprenger, M., and Wernli, H. (2014). A global climatology of stratosphere– troposphere exchange using the ERA-Interim data set from 1979 to 2011, Atmos. Chem. Phys., 14, 913–937, https://doi.org/10.5194/acp-14-913-2014.

Sprenger, M. and Wernli, H. (2003). A northern hemispheric climatology of cross― tropopause exchange for the ERA15 time period (1979–1993), J. Geophys. Res- Atmos., 108.

Stevenson, D. S., Dentener, F. J., Schultz, M. G., Ellingsen, K., Van Noije, T. P. C., Wild, O. et al. (2006). Multimodel ensemble simulations of present― day and near― future tropospheric ozone. Journal of Geophysical Research: Atmospheres, 111(D8).

Williams, R.S., Hegglin, M.I., Kerridge, B.J., Jöckel, P., Latter, B.G. and Plummer, D.A., 2019. Characterising the seasonal and geographical variability in tropospheric ozone, stratospheric influence and recent changes. Atmospheric Chemistry and Physics, 19(6), pp.3589-3620.

Zeng, G., Morgenstern, O., Braesicke, P. and Pyle, J.A., 2010. Impact of stratospheric ozone recovery on tropospheric ozone and its budget. Geophysical Research Letters, 37(9).

Some of our references

McLinden, C. A., Olsen, S. C., Hannegan, B., Wild, O., Prather, M. J., and Sundet, J.: Stratospheric ozone in 3-D models: A simple chemistry and the cross-tropopause flux, J Geophys Res-Atmos, 105, 14653-14665, 2000.

Murphy, D. M., and Fahey, D. W.: An estimate of the flux of stratospheric reactive nitrogen and ozone into the troposphere, Journal of Geophysical Research, 99, 5325-5332, 1994.

Olsen, S. C., McLinden, C. A., and Prather, M. J.: Stratospheric $N_2O$-$NO_y$ system: testing uncertainties in a three-dimensional framework, J. Geophys. Res., 106, 28771-28784, 2001.

Ruiz, D. J., Prather, M. J., Strahan, S. E., Thompson, R. L., Froidevaux, L., and Steenrod, S. D.: How Atmospheric Chemistry and Transport Drive Surface Variability of N2O and CFC-11, J Geophys Res-Atmos, 126, ARTN e2020JD033979
10.1029/2020JD033979, 2021.

Tian, H. Q., Xu, R. T., Canadell, J. G., Thompson, R. L., Winiwarter, W., Suntharalingam, P., Davidson, E. A., Ciais, P., Jackson, R. B., Janssens-Maenhout, G., Prather, M. J., Regnier, P., Pan, N. Q., Pan, S. F., Peters, G. P., Shi, H., Tubiello, F. N., Zaehle, S., Zhou, F., Arneth, A., Battaglia, G., Berthet, S., Bopp, L., Bouwman, A. F., Buitenhuis, E. T., Chang, J. F., Chipperfield, M. P., Dangal, S. R. S., Dlugokencky, E., Elkins, J. W., Eyre, B. D., Fu, B. J., Hall, B., Ito, A., Joos, F., Krummel, P. B., Landolfi, A., Laruelle, G. G., Lauerwald, R., Li, W., Lienert, S., Maavara, T., MacLeod, M., Millet, D. B., Olin, S., Patra, P. K., Prinn, R. G., Raymond, P. A., Ruiz, D. J., van der Werf, G. R., Vuichard, N., Wang, J. J., Weiss, R. F., Wells, K. C., Wilson, C., Yang, J., and Yao, Y. Z.: A comprehensive quantification of global nitrous oxide sources and sinks, Nature, 586, 248-+, 10.1038/s41586-020-2780-0, 2020.

Young, P. J., Naik, V., Fiore, A. M., Gaudel, A., Guo, J., Lin, M. Y., Neu, J. L., Parrish, D. D., Rieder, H. E., Schnell, J. L., Tilmes, S., Wild, O., Zhang, L., Ziemke, J., Brandt, J., Delcloo, A., Doherty, R. M., Geels, C., Hegglin, M. I., Hu, L., Im, U., Kumar, R., Luhar, A., Murray, L., Plummer, D., Rodriguez, J., Saiz-Lopez, A., Schultz, M. G., Woodhouse, M. T., and Zeng, G.: Tropospheric Ozone Assessment Report: Assessment of global-scale model performance for global and regional ozone distributions, variability, and trends, Elementa-Science of the Anthropocene, 6, 2018.

---

## Author Response (AR2)

Although the authors have dismissed most of my comments as being nonsense, they have reworked the paper substantially and improved the writing and structuring in such way that they answered most of my major concerns in the process. In particular, thank you for acknowledging the need for providing a methodology section, which now clarifies a few things for me, including the source of the e90 tropopause (a concept that was known well to me indeed, but not to the general ACP reader who may cross this manuscript, the sole reason I asked for this information to be added). I thus am happy to recommend this manuscript for publication after simply noting the following points.

1) Re: your answer to the following comment (which I don't repeat here due to its unduly length):

"L400-435 [of initial manuscript] Hegglin and Shepherd (2007) have used ACE-FTS O3/N2O correlations in an extensive comparison to a CCM, so should be cited here. This study reveals how sampling issues can affect the interpretation of tracer-tracer correlations using ozone. In particular, the ACE-FTS instrument exhibits a strong sampling bias with unequal sampling of seasons and hemispheres. Undersampling the full correlation space (since your monthly ACE-FTS data will not have sampled all latitudes in your considered latitude range evenly) is likely to impact your results. The differences (or even apparent agreement!) in the slopes between observations and your model may thus be at least partially explained by this sampling bias. A discussion of the limitation of your approach should thus be added."

As any reader of this comment can ascertain themselves, I was not asking for you to cite this paper for the quantification of STE. Instead, I wanted you to merely acknowledge the fact that the ACE-FTS slopes were used for model transport & chemistry evaluation before and that they were shown to be very sensitive to sampling. I noted in fact that your model slopes have changed from one version of the manuscript to the other without justification, an indicator for such sensitivity? Furthermore, anchoring the endpoints of the slopes in the tropospheric value (another addition to your revised manuscript I noted) was another methodological peculiarity highlighted by Hegglin and Shepherd (2007) but not referenced in your (rather informed) use of ACE-FTS data.

Yes, we agree and added a sentence reflecting their work on Line 433.Regarding the change in slopes in the revised paper, our lengthy effort on revisions found an error in how we calculated those slopes. The previous version incorrectly plotted $N_2O$ kept at 320 ppb (with emissions) while the correct version now shows N2OX (no emissions) scaled to a fixed 320 ppb.

_Inserted this Line 433_
.... N2O in the lowermost stratosphere to establish the ratio of the two STE fluxes. **The ACE-FTS O3:N2O slopes were used for model transport & chemistry evaluation (Hegglin and Shepherd, 2007) and found to be very sensitive to satellite sampling, except in the lowermost stratosphere.**

_Inserted this line 446_

Our method described here for deriving the slopes from the ACE-FTS data is slightly different from that of Hegglin and Shepherd (2007; e.g., we do not anchor the tropospheric point), and we have the advantage of a longer record.

2) L714-728 Your statement that CMAM and observations agree seems not correct to me when looking at Hegglin and Shepherd (2007) Figures 4 and 13. These both indicate that N2O and ozone/N2O slopes in the model are too high when compared to the observations. Without a proper evaluation of the CMAM N2O flux estimates, this discussion seems thus on somewhat weak grounds. I realise here that it is not clear how the authors use N2O from Aura-MLS. Since v3.3 should only be used above 100 hPa (see SPARC report no 7, page 115) and while earlier versions may be more useful but still associated with much larger uncertainties in the LMS, this uncertainty should also be properly documented and accounted for in the discussion. Note, again what you exactly do with MLS is not clear to begin with since an appropriate description is lacking.

This section does not evaluate the N$_2$O abundances in the lowermost stratosphere, only the O3:N2O slopes, which we took from H&S Fig 13cd. H&S Figure 4 is absolute abundance vs latitude and this is not related to the STE fluxes. We realize that taking the O3:N2O slopes from their figure, especially considering the large seasonal range, and so we revised the numbers in Table 1 (used in this paragraph) to be a range: e.g., -23±2 instead of the overly precise -23.0. These are now moved into the text. Looking at Table 1, we believe that this is pretty good agreement in modeling the LMS slopes compared to ACE-FTS observations, and we can thus make the comparison of what the STE flux ratios should be.

We apologize for the MLS confusion we created here, and have tried to make clear that Aura-MLS N2O is ONLY for P < 100 hPa to calculate the loss of N2O and hence the 'implied' STE. Thank you also for the constructive notation (Aura-MLS) to avoid confusion with the LMS acronym. Since the CMAM N2O looks "normal" in the mid-stratosphere, we assume they have the near correct lifetime (based on MLS-N2O) or the SPARC lifetime report, e.g., 110-120 yr, and thus have N2O STE fluxes that are "typical".

*Revision Line 553*
For UCI we calculate NH:SH fluxes of O3 (208:182 Tg-O3/yr) and N2O (5.1:6.4 Tg-N/yr). Thus the mole fraction slopes in the lowermost stratosphere should be -23.8 (NH) and -16.6 (SH). Our model O3:N2O slopes are -23.2 (NH) and -17.5 (SH). Given the seasonal variability and scatter in correlation plots (Figure 7), we count this as consistent. For CMAM, the modeled O3:N2O slopes, **-23±2 (NH) and -18±3** (SH) are similar to ours and **also** to the ACE-FTS observations as analyzed by Hegglin and Shepherd (2007), **-22±4** (NH) and **-14±3** (SH), or by us, -19 (NH) and -15 (SH).

*Revision below Line 560*
"CMAM does not report the **implied** STE N2O fluxes **derived from their photochemical loss of N2O**, but their model seems to match observations of N2O in the middle stratosphere, and so we assume that the **Aura-MLS derived** N2O fluxes are a close estimate (12.9 Tg-N/yr). **Note we are using Aura-MLS N2O values here to calculate the photochemical loss, which occurs**

**in the middle to upper stratosphere.**

We do not view this as a critical assessment of CMAM since it involves us combining diagnostics from  two separate publications **and possibly different model simulations,  .**..

*Table revision*

| LMS O3:N2O slope* | -23.2 | -17.5 | | UCI model |
|---|---|---|---|---|
| | -19.4 | -15.3 | | ACE-FTS observations |
| | **-23±2** | **-18±3** | | CMAM model, Fig 13 of (Hegglin & Shepherd, 2007) |
| | **-22±4** | **-14±3** | | ACE-FTS observations, ibid |
| | | | -20.0 | (Murphy & Fahey, 1994) |
| | | | -22.0 | (McLinden et al., 2000) |

STE
note at bottom of table:
* LMS = lowermost stratosphere **only**. For UCI model, months are selected for highest STE (FMAM in NH, SOND in SH, Fig. 1). For CMAM, **the monthly ranges** from their Fig. 13cd **are estimated**. Where no reference is given, the source is this paper.